# Intrinsic disorder within AKAP79 fine-tunes anchored phosphatase activity toward substrates and drug sensitivity

Patrick J Nygren[1], Sohum Mehta[2], Devin K Schweppe[3], Lorene K Langeberg[1], Jennifer L Whiting[1†], Chad R Weisbrod[4], James E Bruce[3], Jin Zhang[2], David Veesler[5], John D Scott[1]*

[1]Department of Pharmacology, Howard Hughes Medical Institute, University of Washington, Seattle, United States; [2]Department of Pharmacology, University of California, San Diego, San Diego, United States; [3]Department of Genome Sciences, University of Washington, Seattle, United States; [4]National High Magnetic Field Laboratory, Florida State University, Tallahassee, United States; [5]Department of Biochemistry, University of Washington, Seattle, United States

**\*For correspondence:**
scottjdw@u.washington.edu

**Present address:** [†]Thermo Fisher Scientific, Austin, United States

**Competing interests:** The authors declare that no competing interests exist.

**Abstract** Scaffolding the calcium/calmodulin-dependent phosphatase 2B (PP2B, calcineurin) focuses and insulates termination of local second messenger responses. Conformational flexibility in regions of intrinsic disorder within A-kinase anchoring protein 79 (AKAP79) delineates PP2B access to phosphoproteins. Structural analysis by negative-stain electron microscopy (EM) reveals an ensemble of dormant AKAP79-PP2B configurations varying in particle length from 160 to 240 Å. A short-linear interaction motif between residues 337–343 of AKAP79 is the sole PP2B-anchoring determinant sustaining these diverse topologies. Activation with Ca2$^+$/calmodulin engages additional interactive surfaces and condenses these conformational variants into a uniform population with mean length 178 ± 17 Å. This includes a Leu-Lys-Ile-Pro sequence (residues 125–128 of AKAP79) that occupies a binding pocket on PP2B utilized by the immunosuppressive drug cyclosporin. Live-cell imaging with fluorescent activity-sensors infers that this region fine-tunes calcium responsiveness and drug sensitivity of the anchored phosphatase.
DOI: https://doi.org/10.7554/eLife.30872.001

## Introduction

Processing information through cell signaling complexes utilizes all four levels of protein structure. Protein-interaction modules delineate the architecture of these macromolecular assemblies by aligning signaling elements in relation to scaffolding components. These structurally distinct protein folds often recognize primary structure determinants embedded within reciprocal binding surfaces (*Scott and Pawson, 2009*). Advances in electron microscopy (EM) emphasize that protein flexibility is another feature that influences how multivalent enzyme complexes adapt to the continually changing intracellular environment (*Saibil, 2000*). For example, intrinsic disorder within regulatory domains of constrained protein kinases guides the phosphorylation of nearby substrates (*Smith et al., 2013*), whereas association of phosphatases with targeting subunits can bias catalytic efficiency toward certain phosphoproteins (*Hendrickx et al., 2009*; *Nygren and Scott, 2015*). Hence, conformational order and disorder act in concert to fine-tune intracellular signaling scaffolds.

Although protein kinases recognize consensus sequence motifs flanking phospho-acceptors (*Yaffe et al., 2001*), compartmentalization via association with anchoring and scaffolding proteins is an equally important determinant of substrate specificity (*Scott et al., 2013*). In parallel, phosphatase-targeting subunits orchestrate where and when dephosphorylation events occur in the cell

**eLife digest** Signaling molecules such as the hormone epinephrine (also known as adrenaline) activate a range of responses inside cells. The responses often involve proteins being chemically modified to change how active they are, which in turn controls specific processes happening inside the cell. One type of modification involves certain enzymes adding or removing molecules known as phosphate groups from specific proteins. For example, an enzyme called PP2B (also known as calcineurin) is able to remove phosphate groups from a variety of proteins.

PP2B plays crucial roles in many different processes in animals including immune responses, nerve cell signaling and heart activity, and is the target of several medicinal drugs that suppress the immune system. Since PP2B plays so many roles in the body, these drugs often have unintended side effects. Therefore, studying how the body regulates this enzyme may help us to understand what causes these side effects.

Previous studies have shown that PP2B is activated by calcium ions, which can act as signals in many different situations inside cells. A protein called AKAP79 anchors PP2B to specific locations in the cell so that it only operates where it is needed. Some evidence suggests that calcium ions affect how AKAP79 and PP2B interact, but it is not known how this works. Nygren et al. investigated how the PP2B enzyme and AKAP79 protein interact inside human cells and in cell-free systems.

The experiments showed that short regions within the AKAP79 protein are responsible for binding to PP2B. These regions and the flexible structure of the entire AKAP79 protein work together to fine-tune how PP2B responds to calcium ions. In the presence of higher levels of calcium ions, another 'auxiliary' region of AKAP79 also binds to PP2B. This auxiliary region binds to a site on the enzyme where an immunosuppressive drug called cyclosporine can also bind. This suggests that AKAP79 binding to PP2B may affect the sensitivity of the PP2B enzyme to cyclosporine.

This study demonstrates that the activity of PP2B can be precisely controlled by interactions with proteins such as AKAP79. Further work on these interactions may help develop more effective drugs that cause fewer side effects in patients.

DOI: https://doi.org/10.7554/eLife.30872.002

(*Ceulemans and Bollen, 2004*). Additionally, anchored kinases and phosphatases often reside within the same macromolecular complex (*Langeberg and Scott, 2015*). Such pre-formed bi-directional signaling units modulate key physiological processes including immune responses, glucose homeostasis, renal water balance and excitatory synaptic transmission (*Clipstone and Crabtree, 1992*; *Hinke et al., 2012*; *Nygren and Scott, 2016*; *Sanderson et al., 2012*; *Whiting et al., 2016*). A unifying molecular principle shared by these wide-ranging physiological responses is that signal termination through targeted phosphatases is the predominant regulatory step. Dephosphorylation resets the system to prime anchored enzyme assemblies for the next pulse or wave of signal. This concept is vividly illustrated in the heart where kinase-phosphatase signaling complexes repetitively respond to synchronized pulses of $Ca^{2+}$ and cAMP during excitation-contraction coupling (*Bers, 2008*; *Lygren and Taskén, 2006*).

We investigate how intrinsic disorder influences the maintenance and mechanics of complexes between the calcium/calmodulin dependent protein phosphatase-2B (PP2B, also called calcineurin or PPP3) and the prototypic anchoring protein AKAP79. Stable interaction with PP2B proceeds through a PxIxIT-like motif that was originally identified in the NFAT transcription factors and the regulator of calcineurin (RCAN), whereas transient contact engages other binding surfaces including LxVP-related sequences (*Dell'Acqua et al., 2002*; *Park et al., 2000*; *Rodríguez et al., 2009*; *Roy and Cyert, 2009*; *Roy et al., 2007*). We show that intrinsic disorder within dormant AKAP79-PP2B assemblies permits a range of extended configurations that are condensed into globular signaling complexes following activation with calcium. Cellular validation for this model is provided by live-cell imaging with fluorescent reporters of anchored PP2B activity.

## Results

### Disorder and short linear motifs within AKAP79

Bioinformatic analyses indicate that AKAP79 consists of defined protein interaction modules linked by regions of intrinsic disorder. Analysis with IUPred (*Dosztányi et al., 2005*) and PONDR (*Li et al., 1999*) predict extended regions of disorder within the first 350 amino acids of AKAP79, whereas the C-terminal portion of the anchoring protein is more ordered (*Figure 1A and B*). This is consistent with evidence showing that the C-terminal region encompasses the PKA-binding helix (blue; *Carr et al., 1991*; *Gold et al., 2006*). The ANCHOR and SLiMPred programs identify regions that are predicted to adopt static conformations upon association with protein binding partners (*Dosztányi et al., 2009*; *Mooney et al., 2012*). These include known binding sites within AKAP79 for protein kinase C/calmodulin (green) and PP2B (red) (*Coghlan et al., 1995*; *Dell'Acqua et al., 2002*; *Faux and Scott, 1997*; *Klauck et al., 1996*; *Oliveria et al., 2007*). Short linear motifs of unknown function were also evident, including a prominent peak at residues 122–136 of the anchoring protein (*Figure 1C and D*, orange). These predictive approaches were a prelude to structural analyses of selected AKAP79 sub-complexes.

His-tagged MBP-AKAP79, GST-PP2B holoenzyme and GST-calmodulin (CaM) were expressed in bacteria and purified by affinity chromatography on the appropriate resins (*Figure 1E*). GST tags were removed by proteolytic cleavage. A macromolecular complex was assembled upon incubation of AKAP79 with molar excesses of PP2B holoenzyme and CaM overnight at 4°C. Separation by size-exclusion chromatography on a Superdex 200 column provided a reconstituted AKAP79 complex containing all protein components as assessed by SDS-PAGE analysis (*Figure 1F and G*, lanes 4–7). Limited degradation of certain protein components occurred during assembly of the macromolecular complex. MS/MS sequencing has previously detected PP2B degradation products of 30 and 45 kDa and there is also an untagged AKAP79 form at ~75 kDa (*Gold et al., 2011*). The apparent molecular mass of this protein complex was 450–550 kDa (*Figure 1F*). However, quantitative analysis with in-line multi-angle light scattering (SEC-MALS) revealed a monodispersed complex of 160 kDa (*Figure 1J*). This corresponds to a 1:1:1:1 ratio of protein components within the AKAP79-PP2B-CaM sub-complex. Further characterization by SDS-PAGE confirmed the presence of each protein component (*Figure 1H*), while analysis by native PAGE detected a macromolecular complex in excess of 480 kDa (*Figure 1I*). Since intrinsically disordered proteins often elute from size-exclusion columns early, we suggest that AKAP79 complexes are disordered.

### Conformational variability in AKAP79/PP2B assemblies

Dormant PP2B is auto-inhibited by a helical segment of its A subunit. Binding of Ca$^{2+}$/CaM relieves inhibition to expose the active site of the phosphatase ([*Li et al., 2011*], *Figure 2A*). An important question is whether tethering to AKAP79 alters the activation of PP2B. Therefore, AKAP79/PP2B holoenzyme complexes formed either in the presence of the calcium chelator EDTA or with excess Ca$^{2+}$/CaM were separated by size-exclusion chromatography (*Figure 2B and C*). Both complexes eluted from the HiLoad Superdex 200 s column with slightly differing retention times. However, although we sometimes observed the active complex (containing CaM) eluting slightly earlier than the dormant complex, the retention time discrepancies were not replicable, due to the fact that these purification steps were carried out to be preparatory rather than analytical. The protein composition of peak fractions (shaded areas) was analyzed by SDS-PAGE (*Figure 2D*). A modified GraFix glutaraldehyde fixation protocol was used to preserve the integrity of each purified complex for subsequent structural analyses (*Kastner et al., 2008*). Migration as a single high-molecular weight band on SDS-PAGE confirmed the stabilization of macromolecular assemblies. These high molecular weight complexes did not have degradation products (*Figure 2D*). Immunoblot analyses confirmed the presence of each protein component in the cross-linked macromolecular assemblies (*Figure 2E–G*). We observed that the anti-PP2B B subunit antibody had some cross reactivity with CaM, which led to a slightly stronger western blot signal in the active complex than in the dormant complex. Samples were applied to EM grids and uranyl formate-stained prior to image acquisition and random conical-tilt analyses (*Figure 2H and I*). Particles of 150–200 Å in diameter were auto picked from micrographs using the DoG (Difference of Gaussians) Picker program (*Voss et al., 2009*).

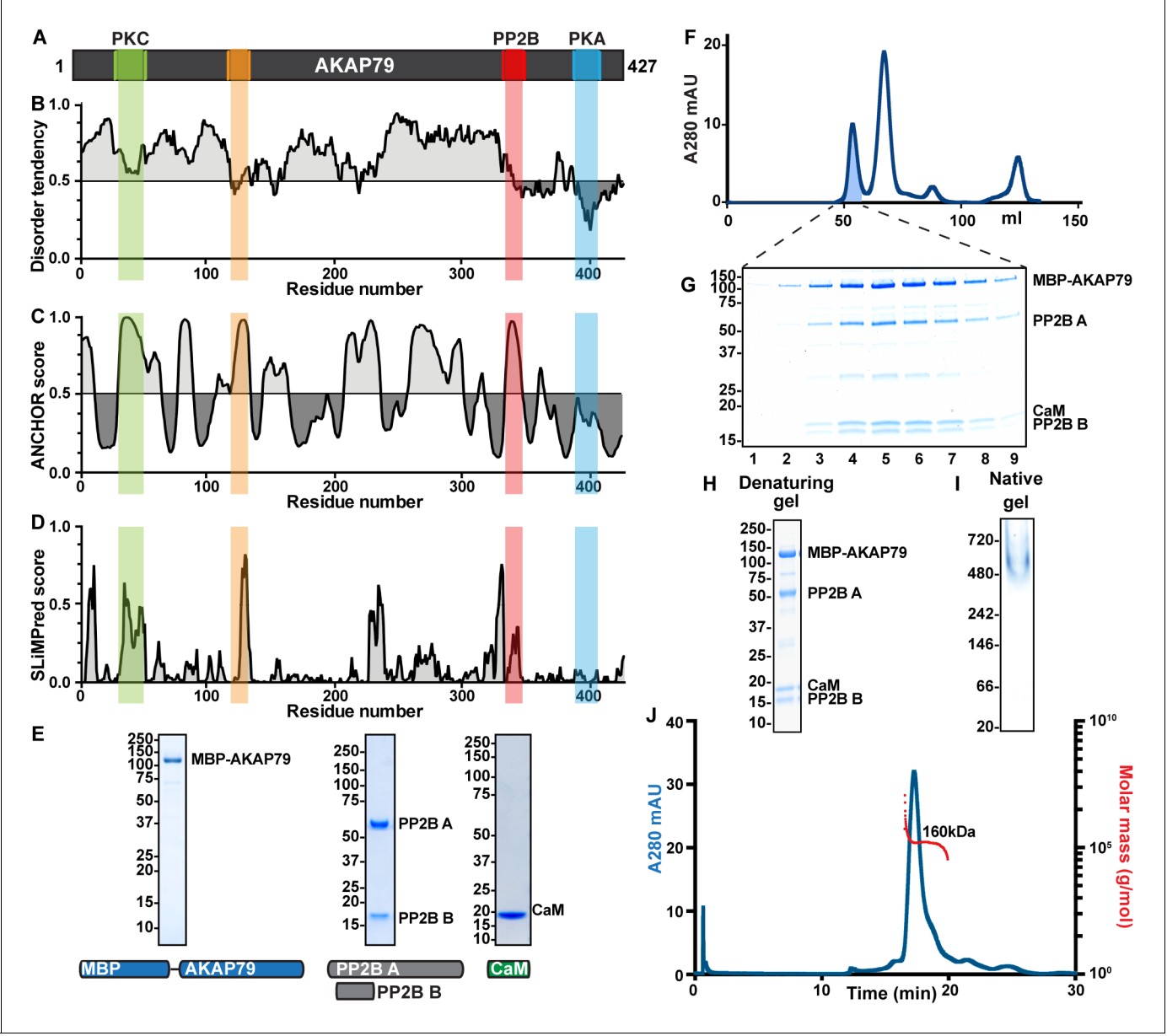

**Figure 1.** Characterizing disorder and short linear motifs in AKAP79 complexes. (**A**) Primary topology of AKAP79, with new and known binding sites notated and shaded. (**B**) IUPred prediction of disordered regions of AKAP79. (**C**) PONDR prediction of disordered regions of AKAP79 (**D**) ANCHOR prediction of short linear motifs in AKAP79. (**E**) SLiMPred prediction of short linear motifs in AKAP79. (**F**) SDS-PAGE gels and constructs used for individual subunits of an MBP-AKAP79/PP2B/CaM complex. (**G**) Gel filtration of a fully assembled AKAP79/PP2B/CaM complex. (**H**). SDS-PAGE gel of fractions pooled for further analysis. (**I**) SEC-MALS of MBP-AKAP79/PP2B/CaM. A280 in blue, molecular weight measurement in red. Insets: denaturing and native gels of the protein complex.

DOI: https://doi.org/10.7554/eLife.30872.003

Random conical-tilt (RCT) images were acquired at 55° angles and paired to cognate un-tilted particles using TiltPicker (*Figure 2H and I*, [*Voss et al., 2009*]).

Reference-free class averaging of dormant AKAP79/PP2B sub-complexes yielded a refined data set of 10,652 elements. The reference-based alignment program SPIDER generated 25 representative classes, each containing ~500–1100 particles (*Figure 3A*). Conformational diversity within these structural classes was immediately evident. To validate this observation we measured the principle axis (length) of each class and weighted this value by the number of particles in the class

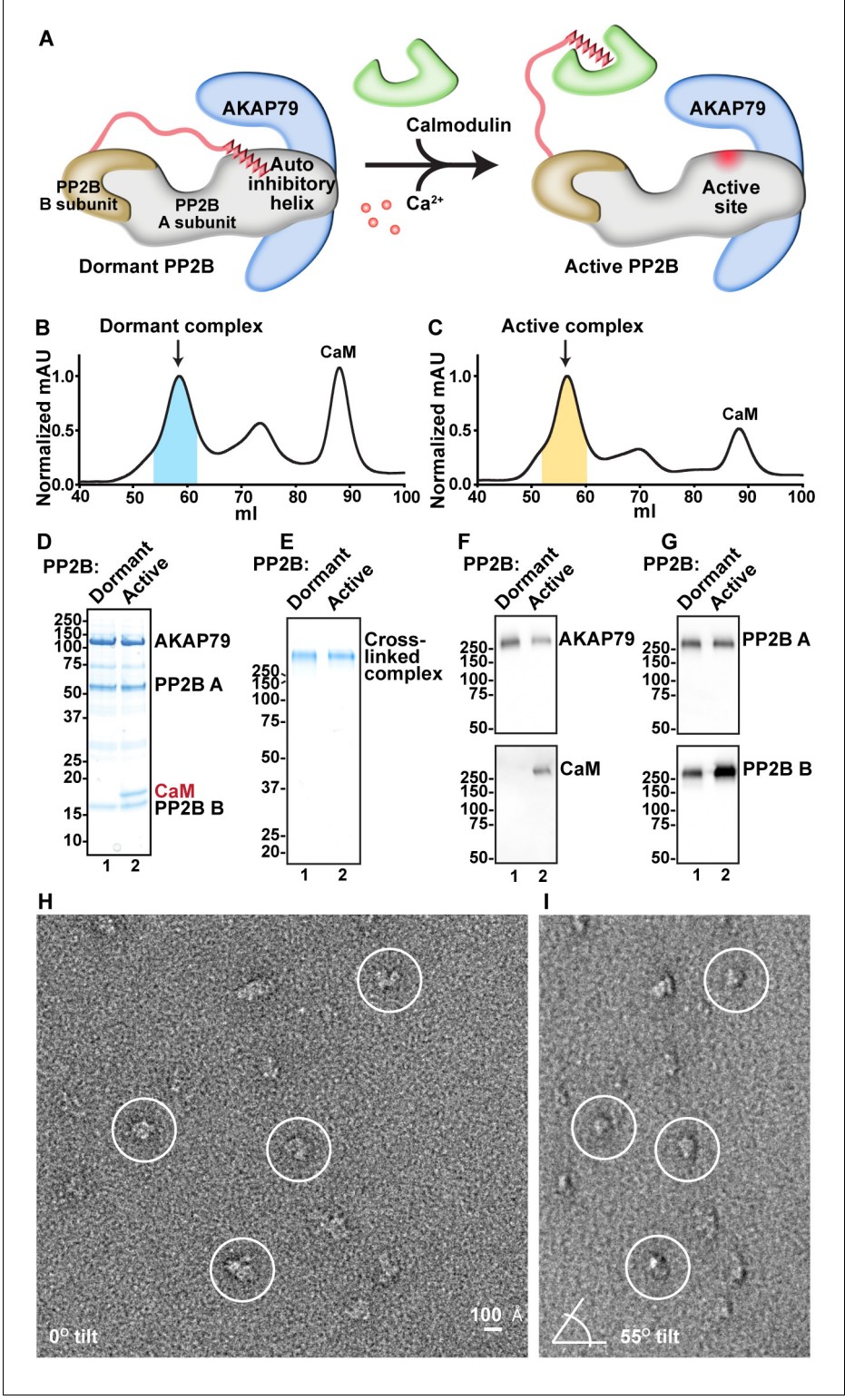

**Figure 2.** Analysis of compositional and conformational changes induced by Ca²⁺. (**A**) Simplified schematic of general PP2B structural rearrangements upon addition of Ca²⁺/CaM. (**B**) SEC UV trace of AKAP79/PP2B/CaM complex obtained in the absence of Ca²⁺. Blue shading indicates fractions analyzed. (**C**) SEC UV trace of AKAP79/PP2B/CaM complex obtained in the presence of Ca²⁺. Yellow shading indicates fractions analyzed. (**D**) SDS-PAGE of pooled peak fractions from dormant and active complexes. (**E**) SDS-PAGE of GraFix purification and crosslinking of dormant and active complexes. (**F**) Western blots for the C-terminal region of AKAP79 and CaM.
*Figure 2 continued on next page*

*Figure 2 continued*

(G) Western blots for PP2B A and B subunits. (H) Example untilted micrograph of the dormant complex. (I) Example tilted micrograph of the dormant complex. Paired particles indicated by circles.
DOI: https://doi.org/10.7554/eLife.30872.004

(*Figure 3B*). Particles segregated into two populations with mean lengths of 184 ± 16 Å (n = 7499) and 228 ± 5 Å (n = 3153, *Figure 3B*, blue). Three-dimensional rendering of selected RCT classes exhibited a range of configurations (*Figure 3C–E*). These models, although of similar width and depth (*Figure 3C–E*, bottom) vary in length from 160 to 240 Å (*Figure 3C–E*, top). Thus, dormant AKAP79-PP2B holoenzyme complexes exist in a variety of flexible and extended topologies.

In contrast, active AKAP79/PP2B/CaM complexes are more uniform. Reference-based alignment of a refined data set of 20,461 elements yielded 25 representative classes that segregated into a single population of active particles with a mean length of 178 ± 17 Å (*Figure 3F and G*, orange). Three-dimensional rendering of representative AKAP79/PP2B/CaM classes reveals more compact and globular complexes (*Figure 3H–J*). Yet both the dormant and the active macromolecular units occupy similar volumes (7.4 × 10$^5$ Å$^3$), when isosurface thresholds are set to match 2-D averages. This implies that additional intermolecular contacts residing within the active AKAP79/PP2B/CaM complex sustain a more globular topology.

Multimodal subunit mapping was used to position individual protein components within a 3D model of the active AKAP79/PP2B/CaM assembly. First, Fab fragments of IgG against the MBP tag localized the N-terminus of AKAP79 in a density at the top of a predominant class average (*Figure 4A–G*). Second, a 1.8 nm nanogold-NiNTA conjugate labeled the His-tag on the C-terminus of the anchoring protein (*Figure 4H–M*). Gold particles decorated the core of the active AKAP79/PP2B/CaM complex (*Figure 4J and M*). Third, Fab fragments against Flag-tagged A subunit of PP2B localized the catalytic subunit of the phosphatase in the lower lobe of the macromolecular assembly (*Figure 4N–R*). This data was compared with unlabeled RCT models and consolidated into a composite color-coded model depicting the topology of an active AKAP79/PP2B/CaM complex processively rotated by 90˚ (*Figure 4S–V* and *Figure 4—video 1*). The predicted location of CaM (blue) is also indicated.

## New intermolecular contacts between AKAP79 and active PP2B

A combined chemical cross-linking and mass spectrometry approach was employed to further define the intermolecular contacts between AKAP79 and the PP2B holoenzyme. Purified dormant and active assemblies were subjected to chemical cross-linking using 10 mM biotin-aspartate proline-PIR n-hydroxyphthalimide (BDP-NHP) at room temperature for 1 hr. After denaturation and trypsin digestion, cross-linked peptides were analyzed by liquid chromatography Mass Spec (LC-MS$^3$). Real time Analysis for Cross-linked peptide Technology (ReACT), an algorithm to identify and assign sequences to cross-linked peptide pairs, charted protein-protein interaction surfaces within these macromolecular assemblies (*Weisbrod et al., 2013*).

Distinct interaction maps were derived for the dormant and active AKAP79/PP2B complexes (*Figure 5A and B*; *Supplementary file 1*). In the dormant complex, a consensus PP2B interaction motif located between residues 337–343 of AKAP79 (PIAIIIT) was the only site of contact with the phosphatase (*Figure 5A* red and *Supplementary file 1*). In contrast, a total of four additional AKAP79-PP2B A subunit interfaces were evident in the active complex (*Figure 5B*, red). These ancillary AKAP-interactive peptides reside within the lower lobe of the catalytic subunit of PP2B (*Figure 5C and D*, green, blue, teal and lime) and are proximal to a previously defined binding surface for the PIAIIIT motif (*Li et al., 2012*). This consolidates the finding that the active AKAP79-PP2B-CaM complex is more compact (*Figure 3*).

ReACT also identified a cross-link between lysine 126 on AKAP79 and lysine 165 on the B subunit of PP2B (*Figure 5B*, orange). This peptide pair was exclusively detected in the active complex. We can therefore infer that amino terminal regions of the anchoring protein contact the B subunit of the active phosphatase holoenzyme (*Figure 5C and D*, orange peptide). A recognizable feature of the AKAP79 peptide is the sequence Leu-Lys-Ile-Pro, which resembles a consensus LxVP phosphatase interaction module (*Rodríguez et al., 2009*; *Sheftic et al., 2016*). Interestingly, this sequence is

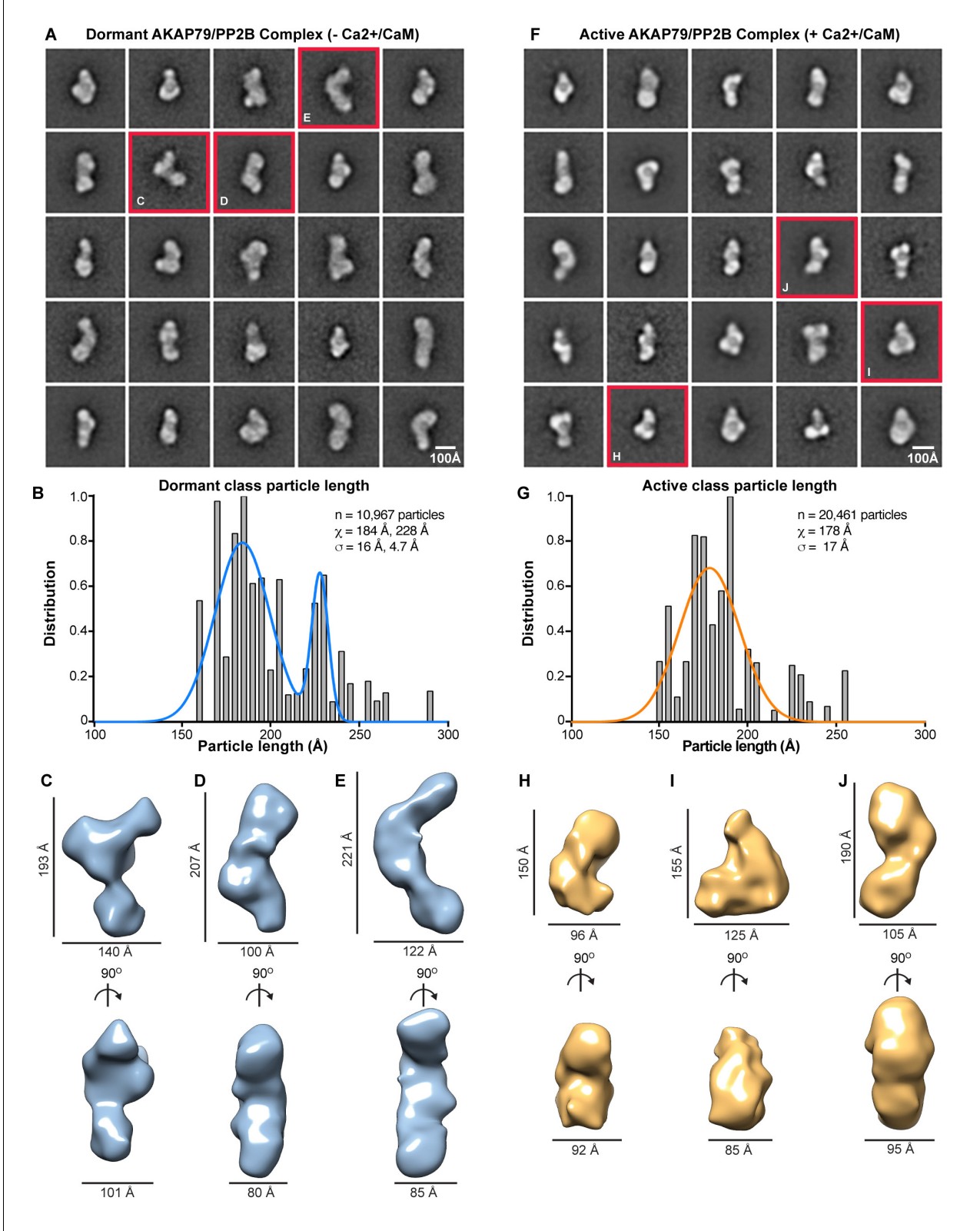

**Figure 3.** EM analysis of AKAP79/PP2B complexes. (**A**) Reference-based class averages of the dormant complex. (**B**) Histogram analysis of dormant class dimensions. Bimodal Gaussian fit in blue. (**C–E**) Sample 3-D models of dormant AKAP79/PP2B illustrating representative states of the complex, notated with dimensions. (**F**) Reference-based class averages of active AKAP79/PP2B/CaM. (**G**) Histogram analysis of active class dimensions. Gaussian fit in orange. (**H–J**) Sample 3-D models of active AKAP79/PP2B/CaM, annotated with dimensions.

*Figure 3 continued on next page*

*Figure 3 continued*

DOI: https://doi.org/10.7554/eLife.30872.005

located within a disordered region of the anchoring protein predicted to form a short linear interaction motif (*Figure 1D and E*, orange bar). Moreover, comparison with a structure of PP2B in complex with the immunosuppressive agent cyclosporin/cyclophilin suggests that this segment of AKAP79 may target the same binding surface as the drug complex (*Figure 5E–F*, [*Jin and Harrison, 2002*]). Molecular modeling predicts that the core Leu-Lys-Ile-Pro (LKIP) motif on AKAP79 slots into this drug-binding pocket on PP2B (*Figure 5G*). Yet, one distinction is that residues downstream of this motif contact PP2B (*Figure 5G*). This is in contrast to conventional PP2B interacting partners that

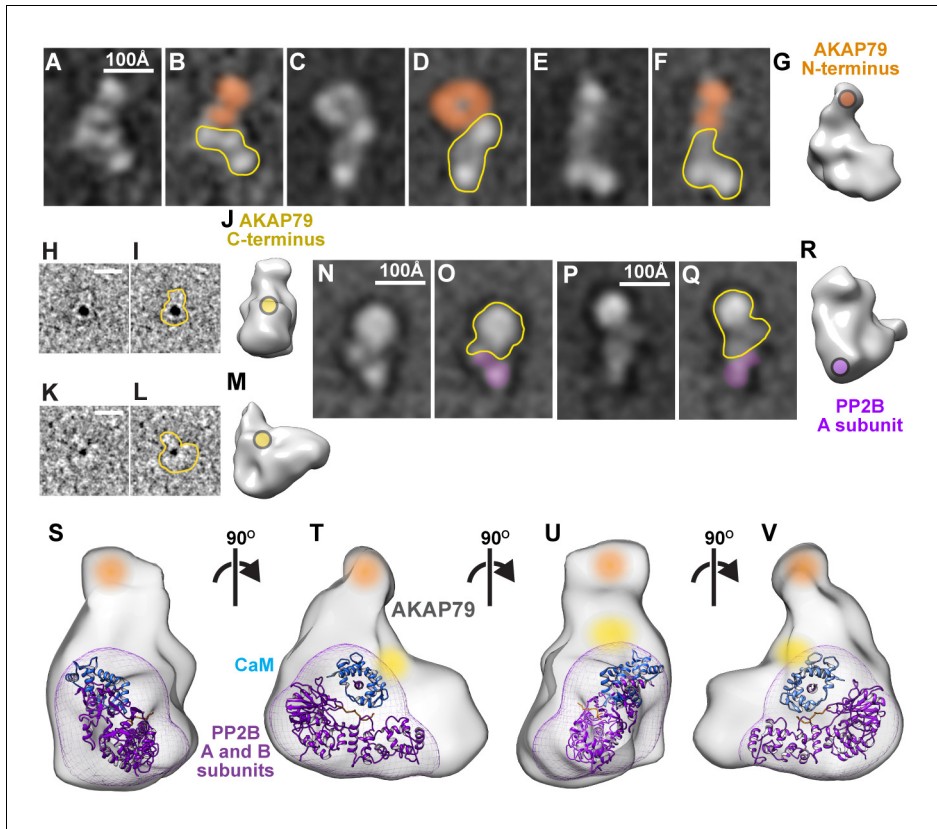

**Figure 4.** Subunit mapping of AKAP79/PP2B/CaM complexes. (**A–F**) Reference-free class averages of AKAP79/PP2B/CaM assemblies complexed with anti-MBP-AKAP79 Fab fragments. Density corresponding to anti-MBP Fab fragment notated with orange. AKAP79/PP2B/CaM complex outlined in yellow (**G**) 3-dimensional model with anti-MBP Fab binding site indicated in orange. (**H, K**) Representative Ni-NTA gold-labeled single particle images of AKAP79/PP2B/CaM, 10 Å low-pass filtered for visualization. (**I, L**) Gold-labeled particle outlines indicated in yellow. (**J, M**) 3-dimensional models with gold-labeled 10x-His tag indicated in yellow. (**N–Q**) Reference-free class averages of AKAP79/PP2B/CaM assemblies complexed with anti-Flag-PP2B Fab fragments. Density corresponding to anti-MBP Fab fragment denoted with purple. AKAP79/PP2B/CaM complex outlined in yellow (**R**) 3-dimensional model with anti-Flag-PP2B Fab binding site indicated in purple. (**S–V**) 3-dimensional model of active AKAP79/PP2B/CaM complex with subunit labels notated in color and crystal structures of a PP2B (purple)/CaM (blue, PDB: 1YR5) complex docked within the density, views rotated successively around a vertical axis by 90°. See also *Figure 4—video 1*.

DOI: https://doi.org/10.7554/eLife.30872.006

The following video is available for figure 4:

**Figure 4—video 1.** 3D RCT model with subunit crystal structures placed within EM model based on labeling experiments.

DOI: https://doi.org/10.7554/eLife.30872.007

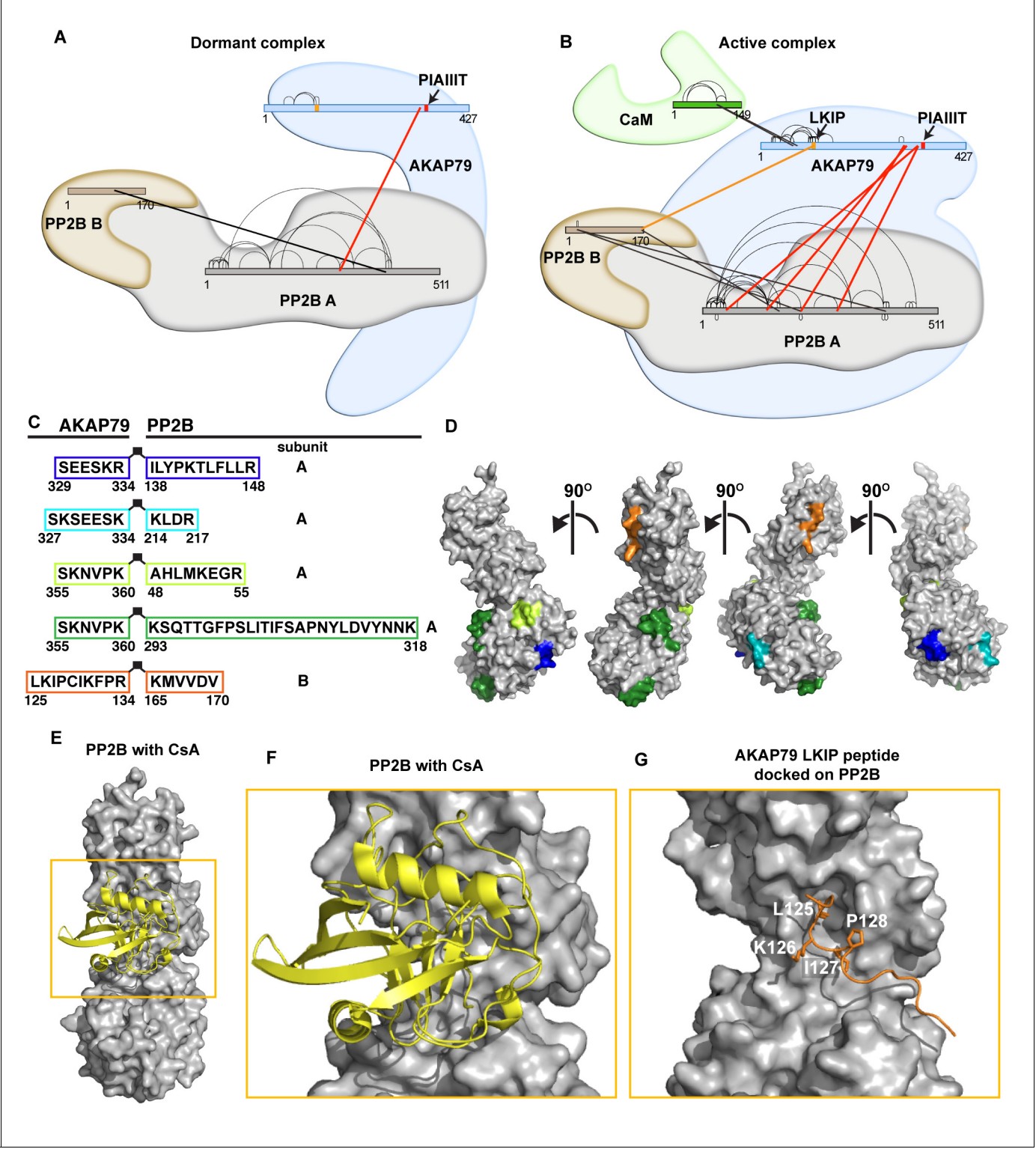

**Figure 5.** Crosslinking/mass-spectrometry of AKAP79/PP2B interactions. (**A**) Crosslink map of residues observed in dormant complexes. Thick lines indicate intermolecular crosslinks, and red lines indicate crosslinks from PP2B to AKAP79. See also *Supplementary file 1*. (**B**) Crosslink map of residues that were observed in active complexes conditions. Thick lines indicate intermolecular crosslinks, and red lines indicate crosslinks from PP2B to AKAP79. Orange line indicates selected crosslink from AKAP79-K126 to PP2B-K165. (**C**) Table of AKAP79-PP2B crosslinked peptides observed in the active complex, color-coded. (**D**) Cognate binding surfaces for these peptides mapped onto the PP2B surface (gray), color-coded with the same scheme as
*Figure 5 continued on next page*

*Figure 5 continued*

the previous panel (PDB: 5SVE). (**E**) PP2B (gray) with cyclosporin/cyclophilin complex (yellow) (PDB: 1MF8). (**F**) Inset showing the immunophilin binding site. (**G**) Rosetta FlexPepDock model of AKAP79 crosslinked peptide (orange) complexed with PP2B (gray).

DOI: https://doi.org/10.7554/eLife.30872.008

utilize determinants upstream of the LxVP motif to enhance the binding affinity for targeting subunits (*Sheftic et al., 2016*).

## The LKIP interaction motif on AKAP79

In vitro biochemical approaches were used to further characterize the new PP2B binding surface on AKAP79. Solid-phase spot-arrays of 20-mer peptides spanning the entire anchoring protein (each displaced by three residues) were screened for interaction with PP2B (*Figure 6A and B*). Analyses were carried out in the presence of $Ca^{2+}$/CaM or EDTA to determine which interactive surfaces were calcium dependent. Residues 122 to 136 of AKAP79 formed a $Ca^{2+}$ responsive PP2B-binding region (*Figure 6B*). Notably, determinants flanking the core Leu-Lys-Ile-Pro quadrapeptide (residues 125–128; LKIP) contributed to $Ca^{2+}$-dependent phosphatase anchoring, since pretreatment with EDTA abolished all protein-protein interactions (*Figure 6B*). Control arrays confirmed tonic PP2B interaction through the well-defined PIAIIIT motif (*Figure 6C*, [*Dell'Acqua et al., 2002*; *Li et al., 2012*; *Oliveria et al., 2007*]). Taken together these studies define a region of AKAP79 between residues 122 and 136 that interfaces with the B subunit of the active PP2B holoenzyme.

Next, GST pull-downs with purified PP2B established whether the LKIP motif is a principle binding-determinant for active PP2B. Purified fragments encompassing residues 1–153, 154–296, and 297–427 of AKAP79 were screened for interaction with the phosphatase (*Figure 6D*). Immunoblot analysis demonstrated that the N-terminal fragment bound to the phosphatase only in the presence of $Ca^{2+}$/CaM (*Figure 6D*, top panel, lane 2). Protein-protein interaction was abolished in the presence of an LxVP competitor peptide derived from the transcription factor NFAT (*Figure 6D*, top panel, lane 4; [*Park et al., 2000*]). Binding was also lost upon chelation of calcium (*Figure 6D*, top panel, lane 3). Likewise, PP2B binding was abolished when experiments were repeated with a mutant N-terminal fragment of AKAP79 that lacked the LKIP motif (*Figure 6D*, top panel lanes 5–8). Control experiments confirmed tonic PP2B binding though the PIAIIIT motif located in the C-terminal fragment of the anchoring protein (*Figure 6D*, top panel lanes 13–16). The middle fragment does not interact with the phosphatase under any conditions (*Figure 6D*, top panel lanes 9–12). Ponceau staining served as a loading control (*Figure 6D*, bottom panel). In parallel, phosphatase activity measurements using the small molecule substrate (diFMUP) demonstrated that AKAP79 associated-PP2B retained full activity (*Figure 6—figure supplement 1*).

AlphaScreen assays were used to measure the binding affinity of PP2B for fragments of the anchoring protein. In order to calculate Kd values, untagged PP2B and $Ca^{2+}$/CaM were used as competitors of AKAP79-phosphatase complexes. An N-terminal fragment of AKAP79 encompassing the LKIP motif bound PP2B with a Kd of 81 μM (n = 4) as compared to binding affinities of 5.6 μM and 12 μM for the PIAIIIT region and the full-length AKAP respectively (both n = 4; *Figure 6E*). These measurements further indicate that the PIAIIIT motif is the primary determinant for PP2B anchoring, whereas the LKIP region provides an auxiliary interaction.

Mechanistic and crystallographic analysis of PP2B has identified two aromatic amino acids (W352 and F356) that form a binding pocket for LxVP motifs (*Figure 6F*) (*Rodríguez et al., 2009*). When both residues were substituted with alanine (W352A/F356A), we noted that this mutant had reduced binding to the N-terminal fragment of AKAP79 in GST pulldown assays (*Figure 6G*, top panel; lane 6). Control experiments confirmed that the AKAP79 fragments encompassing the PIAIIIT motif retained the ability to bind wildtype and mutant PP2B (*Figure 6G* top panel, lanes 7 and 8). Together, these results indicate that the LKIP region on AKAP79 utilizes a similar mode of interaction analogous, but not identical to LxVP motifs. A recent study has suggested that residues immediately upstream of the LxVP core motif (−1 and −2 positions) contribute to PP2B-selective binding (*Sheftic et al., 2016*). One clear distinction in the AKAP79/PP2B interface is that the anchoring protein utilizes distal PP2B binding determinants both upstream and downstream of the LKIP motif.

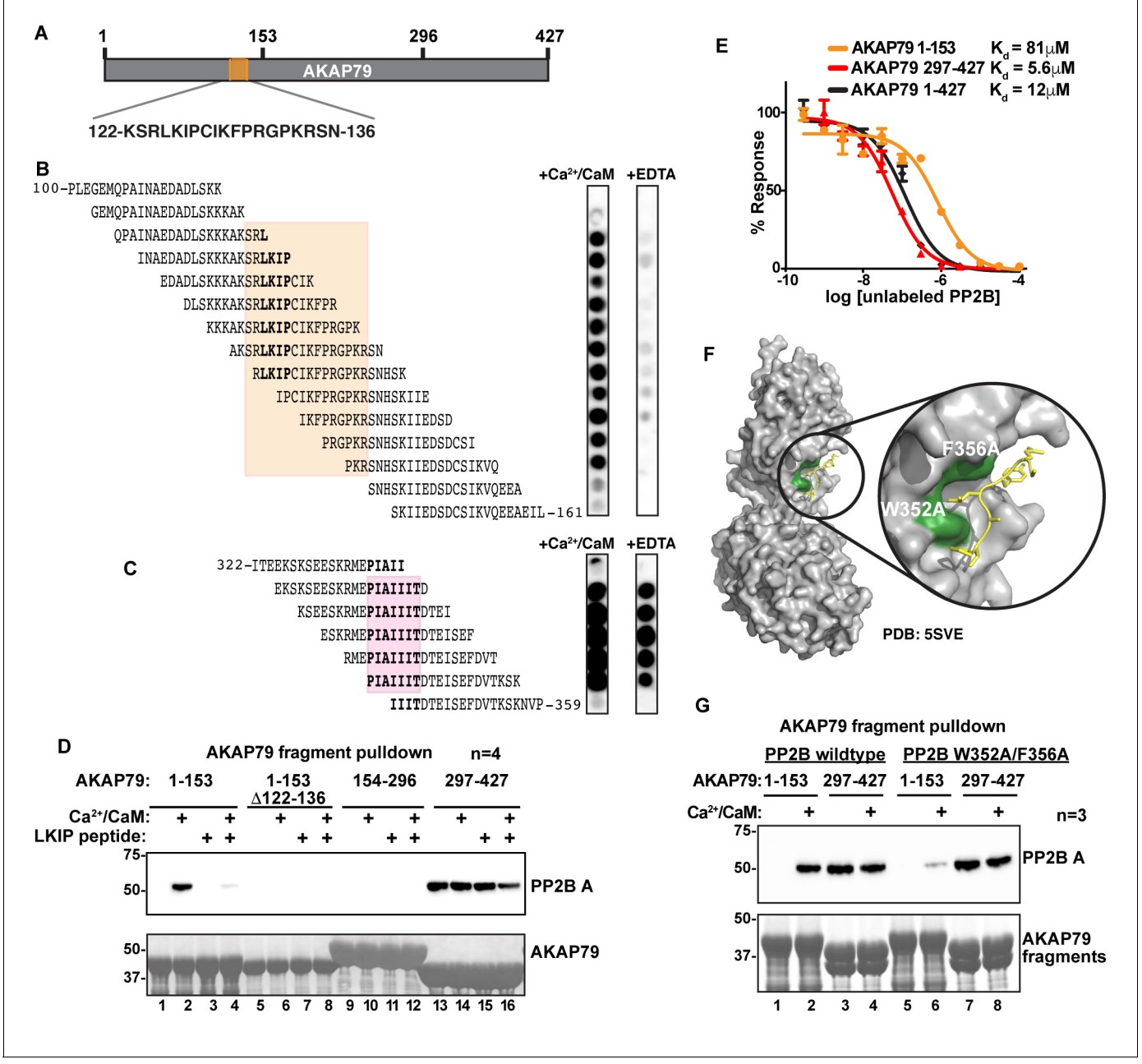

**Figure 6.** Mapping and characterization of the LKIP binding interfaces for AKAP79 and PP2B. (A) Fragments of AKAP79 used in subsequent experiments. (B) PP2B overlay of synthesized peptides in the LKIP region ± Ca$^{2+}$/CaM. Shading indicates sequences shared by Ca2+/CaM-dependent binding peptides. (C) PP2B overlay of synthesized peptides in the PIAIIIT region ± Ca$^{2+}$/CaM. Shading indicates sequences shared PP2B-binding peptides. (D) GST-AKAP79 fragment pulldowns of Flag-PP2B in the presence of Ca$^{2+}$/CaM, and competition with 200 μM NFAT peptide. Lanes 5–8 show that deletion of AKAP79 122–136 abolishes binding. See also *Figure 6—figure supplement 1*. (E) AlphaScreen competition assay to calculate Kd values for interactions between PP2B and full-length AKAP79 (green), or fragments of AKAP 79 (C – blue, N – orange). Data are represented as mean ±SEM. (F) Structural model showing PP2B (gray) with the mutated LxVP binding pocket residues (green) interacting with an LxVP motif (yellow) (PDB: 5SVE). (G) Mutations were made to PP2B that are predicted to abolish the LxVP binding pocket. GST-pulldown assays were used to test mutant PP2B binding to the N-terminal fragment of AKAP79.

DOI: https://doi.org/10.7554/eLife.30872.009

The following figure supplement is available for figure 6:

**Figure supplement 1.** DiFMUP phosphatase activity assay.

DOI: https://doi.org/10.7554/eLife.30872.010

## The LKIP motif controls PP2B sensitivity to calcium/calmodulin

A FRET biosensor of PP2B activity towards anchored substrates was used to investigate dynamic and bipartite interaction with AKAP79 inside cells (*Mehta et al., 2014*; *Mehta and Zhang, 2014*). We reengineered the original calcineurin activity reporter (CaNAR) to utilize AKAP79 as the sole binding partner for PP2B. CaNAR reporter variants lacking the LKIP motif (AKAP79ΔL-CaNAR); the PIAIIIT site (AKAP79ΔP-CaNAR) and a double mutant (AKAP79ΔLΔP-CaNAR) were also generated. Characterization included expression in HEK293 cells to confirm membrane targeting of each CaNAR construct (*Figure 7B*) and western blot analysis of AKAP79 immune complexes to detect co-precipitation of the phosphatase (*Figure 7C*). These latter studies detected co-fractionation of PP2B only with AKAP79-CaNAR and AKAP79ΔL-CaNAR (*Figure 7C*, top panel, quantification in *Figure 7D*). Treatment with 1 μM ionomycin, which increases calcium permeability, augmented detection of PP2B (*Figure 7C*, top panel, lanes 7 and 8). Thus, engaging the LKIP motif favors a stable interaction with AKAP79.

Live-cell imaging was conducted over a time course of 12 min. Changes in YFP/CFP FRET ratio served as an index of PP2B activity (*Figure 7E and F*). A sustained rise in FRET was evident upon stimulation with 1 μM ionomycin in cells expressing AKAP79-CaNAR (*Figure 7E*, grey, n = 26). Qualitatively similar FRET responses were observed with the AKAP79ΔL-CaNAR reporter although the maximal effect was slightly reduced (*Figure 7E and F*, orange, n = 31). FRET responses were negligible in cells expressing AKAP79ΔP-CaNAR that lacks the PIAIIIT motif (*Figure 7E and F*, red, n = 14). Maximal FRET responses from wild-type and AKAP79ΔL-CaNAR constructs were not significantly different (*Figure 7F*), yet rates of anchored PP2B activation varied. The AKAP79ΔL-CaNAR variant reached half maximal response in 2.41 ± 0.24 min as compared to 3.58 ± 0.20 min for the full-length reporter (*Figure 7G*). This change in kinetics did not depend on the expression level of the CaNAR reporter as determined by linear regression analysis of the data (R-squared values of less than 0.1, *Figure 7—figure supplement 1*). We hypothesized that these differences are a consequence of increased sensitivity to calcium in the AKAP79ΔL-CaNAR variant. Accordingly, when experiments were repeated with a ten-fold lower concentration of ionomycin (100 nM), the AKAP79ΔL-CaNAR reporter exhibited statistically higher maximal FRET responses than the native reporter (*Figure 7H–J*). Control imaging with the genetically encoded calcium indicator RCaMP confirmed that total calcium responses were equivalent in the presence of both modified CaNAR reporters (*Figure 7—figure supplement 1*). Collectively, these results indicate that the LKIP motif serves to desensitize PP2B towards low levels of calcium, while still allowing the anchored phosphatase to maintain maximal activity when calcium concentrations are elevated. Our modified CaNAR fusion reporter does not contain PP2B anchoring motifs (LxVP or PxIxIT), therefore, our results suggest that the decrease is not due to competition for LxVP motifs. Rather, it is likely that upon Ca2+/CaM influx, engagement of the LKIP motif on AKAP79 stabilizes a conformation of PP2B that does not relieve auto-inhibition as readily as non-anchored phosphatase. Thus, we propose that the LKIP motif in AKAP79 serves to fine-tune the calcium sensitivity of anchored and active PP2B.

Finally, we investigated whether interaction with AKAP79 affects the sensitivity of PP2B to immunosuppressant drugs. Rationale for this experiment was provided by evidence that the LKIP region of AKAP79 binds at the immunophilin-drug binding site on PP2B (*Figure 5E–G*). Competition experiments performed with cyclosporin A/cyclophilin A cocktails (2 μM) diminished binding of PP2B to the N-terminal region of AKAP79 as assessed by western blot (*Figure 8A–B*; top panel, lane 4). In contrast, the immunosuppressant drug had no effect on PP2B binding to AKAP79 fragments encompassing the PIAIIIT sequence (*Figure 8A–B*, top panel, lanes 5–8). Coomassie blue staining of AKAP79 fragments served as a loading control (*Figure 8A*, bottom panel). Importantly, these results indicate that the LKIP motif and flanking determinants on AKAP79 not only modulate calcium responsive PP2B anchoring, but also may influence the clinical action of immunosuppressant drugs.

## Discussion

Although it is well established that anchored phosphatases and kinases modulate local signaling events, much less is known about structural and mechanistic features that contribute to this process. Using a combined negative-stain EM and chemical cross-linking strategy we discover that conformational flexibility within AKAP79-PP2B complexes is a fundamental element of their molecular action. A key facet of this emerging concept is that anchored signaling enzymes are not only tethered to

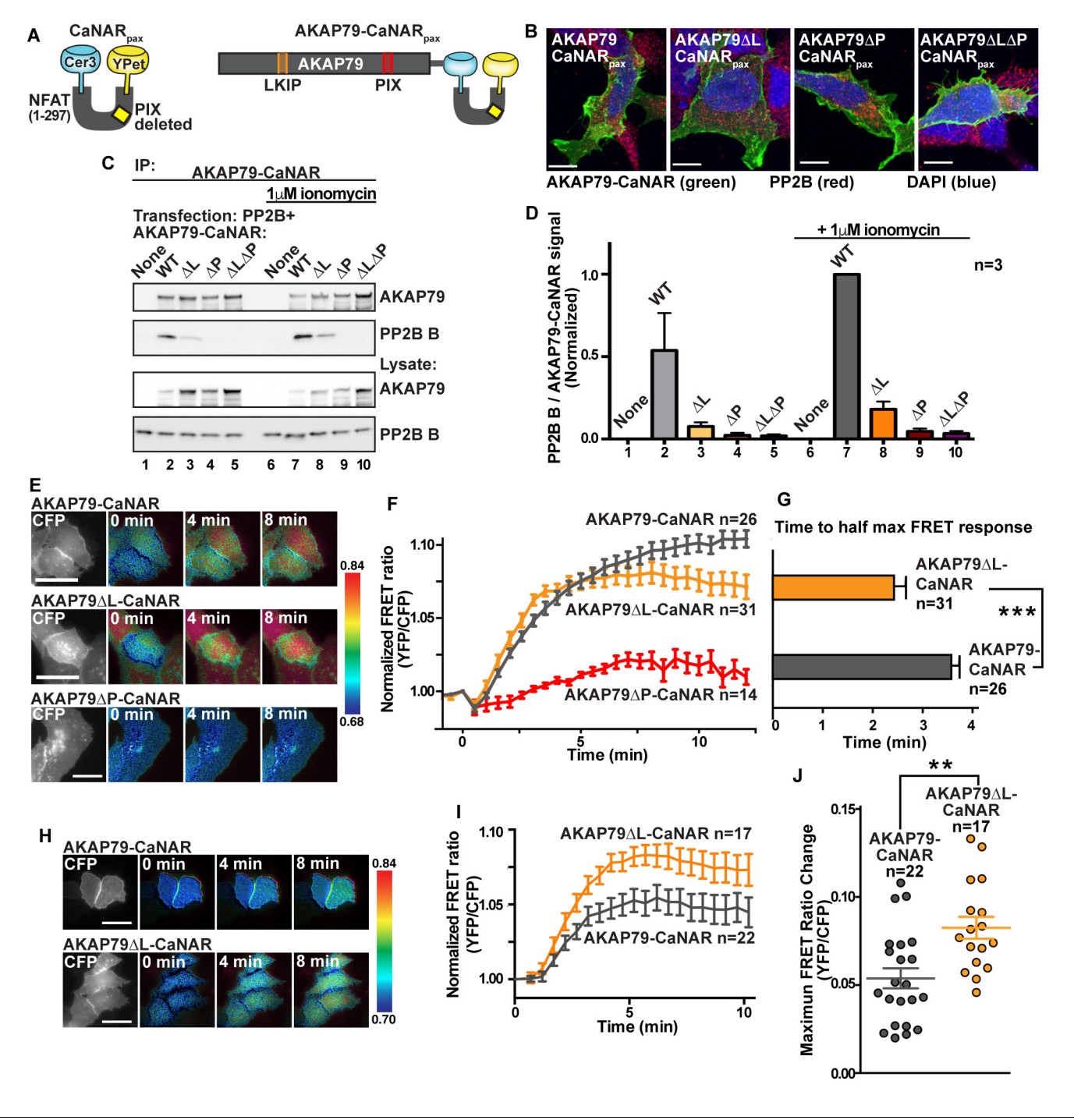

**Figure 7.** Fine tuning of PP2B sensitivity towards physiological ranges of calcium by AKAP79. (**A**) Schematic of the CaNAR reporter with an ablated PxIxIT motif, fused in tandem with full-length AKAP79. (**B**) Confocal images of AKAP79-CaNAR variants (green), PP2B (red) and DAPI (blue) showing proper plasma membrane localization of various mutants. (**C**). Co-immunoprecipitation of PP2B with AKAP79-CaNAR variants under basal (lanes 1–5) and ionomycin treated (lanes 6–10) conditions. (**D**) Quantification of western blot signals of the ratio of co-immunoprecipitated PP2B B subunit to AKAP79-CaNAR variants, normalized to lane 7. Data are represented as mean ±SEM. (**E**) Representative frames of FRET movies comparing wild-type AKAP79-CaNAR with AKAP79ΔL-CaNAR and AKAP79ΔP-CaNAR signals. Colors are representative of the ratio value as described by the key to the right of each row. See also *Figure 7—videos 1–3*. (**F**) Time course of FRET ratio signal upon stimulation with 1 µM ionomycin at t = 0. (**G**) Time to half-max FRET signal after treatment with 1 µM ionomycin. Data are represented as mean ±SEM. (*** indicates p<0.001, unpaired t-test). (**H**) Representative frames of FRET movies comparing wild-type AKAP79-CaNAR with AKAP79ΔL-CaNAR signals after treatment with lower ionomycin concentrations (100 nM). See also *Figure 7—videos 4* and *5*. (**I**) Time course of FRET ratio signal upon stimulation with 100 nM ionomycin at t = 0. (**J**) Treatment of cells

*Figure 7 continued on next page*

*Figure 7 continued*

with 100 nM ionomycin reveals differences in the max response of each variant. Data are represented as mean ±SEM. (** indicates p<0.01, unpaired t-test). See also *Figure 7—figure supplement 1*.

DOI: https://doi.org/10.7554/eLife.30872.011

The following video and figure supplement are available for figure 7:

**Figure supplement 1.** RCaMP measurements of calcium.

DOI: https://doi.org/10.7554/eLife.30872.012

**Figure 7—video 1.** Representative response of wild-type AKAP79-CaNAR to 1 μM ionomycin.

DOI: https://doi.org/10.7554/eLife.30872.013

**Figure 7—video 2.** Representative response of AKAP79ΔL-CaNAR to 1 μM ionomycin.

DOI: https://doi.org/10.7554/eLife.30872.014

**Figure 7—video 3.** Representative response of AKAP79ΔP-CaNAR to 1 μM ionomycin.

DOI: https://doi.org/10.7554/eLife.30872.015

**Figure 7—video 4.** Representative response of wild-type AKAP79-CaNAR to 100 nM ionomycin.

DOI: https://doi.org/10.7554/eLife.30872.016

**Figure 7—video 5.** Representative response of wild-type AKAP79ΔL-CaNAR to 100 nM ionomycin.

DOI: https://doi.org/10.7554/eLife.30872.017

subcellular organelles, but intrinsic disorder within ancillary targeting components confers a dynamic range of enzyme action. For example, flexible linker regions within each regulatory subunit (RII) of anchored Protein Kinase A (PKA) confer a ~200 Å radius of motion to the associated catalytic subunits (*Smith et al., 2013*). More recently we have shown that these anchored PKA holoenzymes remain intact, active and proximal to substrates within these newly defined signaling islands (*Smith et al., 2017*). In other cellular contexts, association with AKAPs exerts allosteric effects that protect PKC isoforms from ATP-competitive inhibitors (*Hoshi et al., 2010*). This may explain the mixed success of clinical trials with certain PKC inhibitors and how phosphorylation-dependent regulation of ion channels in the sympathetic nervous system is refractory to these small-molecule antagonists of PKC (*Bosma and Hille, 1989*; *Mochly-Rosen et al., 2012*). These observations indicate that protein-protein interactions contribute to the substrate preference of these often-promiscuous enzymes, and in a broader context point towards intrinsic disorder and conformational flexibility as inherent specificity determinants of anchored enzyme scaffolds.

Our findings expand on this concept, showing that unstructured regions within the prototypic anchoring protein AKAP79 contribute to targeting and modulation of protein phosphatase 2B. We have discovered short linear motifs (SLiMS) interspersed between regions of flexibility within AKAP79 that tailor phosphatase targeting interactions to permit local and dynamic control of protein dephosphorylation. Another new concept uncovered by our EM analyses is that phosphatase activation induces a topological reorganization of AKAP79-PP2B assemblies. We base this conclusion on single-particle EM evidence in *Figure 3* showing that upon addition of Ca$^{2+}$/calmodulin, dormant AKAP79-PP2B assemblies transition from an assortment of extended multi-lobal conformations into more globular structures. The crosslinking studies in *Figure 5* implicate the PIAIIIT motif at residues 337–343 of AKAP79 as a stable PP2B-anchoring determinant that forms the conformational core for dormant assemblies. However, the bi-modal distribution in particle sizes highlighted in *Figure 3C* may reflect the differential deployment of other SLiMs that act synergistically with the PIAIIIT anchoring motif (*Grigoriu et al., 2013*; *Rodríguez et al., 2009*; *Roy et al., 2007*). This configuration is radically altered upon activation with Ca$^{2+}$/CaM as the aforementioned topological variants consolidate into a single population with a mean particle length of 178 Å. Hence, additional SLiMs are engaged within the active assembly to drive the formation of a more compact, but fully active phosphatase-anchoring protein unit.

Crosslinking data presented in *Figure 5B* supports this finding, identifying four additional interactive surfaces between AKAP79 and the lower lobe of the catalytic core of the active phosphatase. In this study we focused on an 11-amino acid calcium-dependent interaction surface within AKAP79 that contacts the B subunit of PP2B. Although peptide array mapping in *Figure 6* identifies binding determinants throughout this region, the predominant contact site is a Leu-Lys-Ile-Pro sequence located between residues 125–128 of the anchoring protein. There are several interesting features of this interactive surface. This sequence closely resembles an LxVP targeting motif that has been

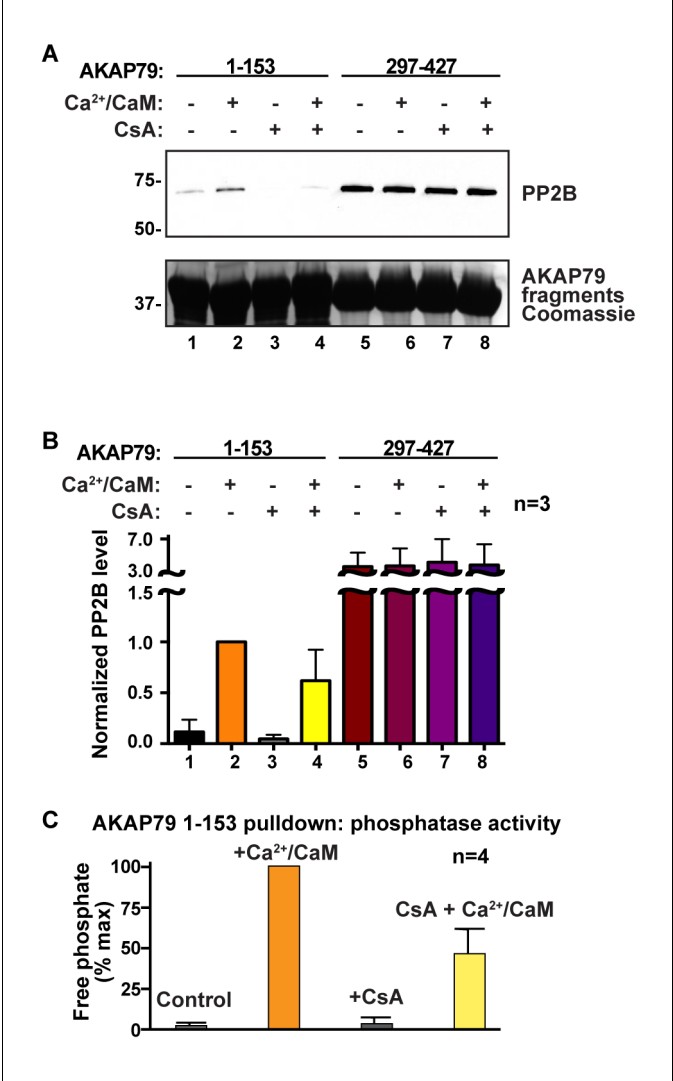

**Figure 8.** AKAP79 and cyclosporin bind to a common surface on PP2B. (**A**) GST-pulldown experiments using N- and C- fragments of AKAP79 and testing for competition using cyclosporin/cyclophilin complexes, n = 3. (**B**) Quantification of PP2B signals in GST-pulldowns, normalized to lane 2, n = 3. Data are represented as mean ± SEM. (**C**) Phosphatase activity assay on samples from lanes 1–4 using a phosphopeptide substrate, n = 4. Data are represented as mean ± SEM.
DOI: https://doi.org/10.7554/eLife.30872.018

previously implicated in a variety of PP2B actions (*Escolano et al., 2014*; *Nygren and Scott, 2016*; *Sheftic et al., 2016*). The LxVP motif was originally identified as a rudimentary PP2B substrate recognition site that enhances dephosphorylation of the RII subunit of PKA (*Blumenthal et al., 1986*). Subsequently, LxVP motifs were shown to augment PP2B binding and dephosphorylation of the NFAT transcription factors (*Grigoriu et al., 2013*; *Rodríguez et al., 2009*). More recently, a variation on this theme was reported in the MAP kinase scaffolding protein KSR2 where an LxVP motif is utilized in conjunction with intrinsic disorder to ensure efficient dephosphorylation of serine and threonine residues at sites distal to the PP2B binding site (*Brennan et al., 2011*; *Dougherty et al., 2009*). A parallel situation may arise in AKAP79 macromolecular assemblies, as flexibility surrounding the PP2B-binding regions of the anchoring protein may provide a means to extend the range of anchored phosphatase action.

Scrutiny of type I phosphatase (PP1)-targeting subunit co-crystal structures shows that the LKIP motif on AKAP79 is analogous to the MyPhonE or SILK motifs (*Hendrickx et al., 2009*). These

secondary binding determinants act in concert with RVxF targeting elements to manage the PP1 catalytic subunit in a variety of subcellular compartments (*Egloff et al., 1997*; *Roy and Cyert, 2009*). Live cell imaging with AKAP79-CaNAR FRET reporters (*Figure 7D and E*) expand this concept by showing that a protein-protein interaction through LKIP sets distance parameters for anchored phosphatase activity that are contingent on the prior association of the PP2B A subunit with the PIAIIIT motif. Hence, the cooperative action of multiple enzyme-binding surfaces, along with the malleability of the AKAP, creates an elastic and adaptive signaling environment that permits the phosphatase to reach nearby substrates.

Molecular docking using the Rosetta FlexPepDock program uncovers another intriguing feature of the LKIP motif on AKAP79. This predictive approach reveals that the LKIP sequence recognizes a binding surface on PP2B utilized by the immunosuppressant drug cyclosporin (*Figure 5E–G*; [*Jin and Harrison, 2002*]). This is borne out by competition experiments presented in *Figure 7H* showing that excess cyclosporin antagonizes AKAP79-phosphatase interaction. This finding may have implications for how this signaling complex operates in a variety of pathological contexts. For example, tissue-specific ablation of PP2B from β-cells generates a diabetic state that includes defects in NFAT transcriptional signaling, diminished insulin biosynthesis and a loss of islet mass (*Bernal-Mizrachi et al., 2004*; *Heit et al., 2006*). Likewise, organ transplant patients who continually receive the PP2B inhibitor drugs cyclosporin and FK506 as immunosuppressives often experience increased blood glucose levels and hypertension as hallmarks of a clinical syndrome known as new-onset diabetes after transplant (*Heisel et al., 2004*). AKAP150 knockin mice that lack the PIAIIIT motif and incorrectly target the active phosphatase display improved glucose handling and heightened insulin sensitivity (*Hinke et al., 2012*). This argues that mislocalization of PP2B is the predominant molecular event underlying these advantageous, albeit tissue-specific, metabolic phenotypes that counteract the symptoms of type II diabetes. Thus, small molecules or peptidomimetics that perturb PP2B tethering, but do not affect the catalytic activity, may boost insulin sensitivity (*Matsoukas et al., 2015*). This theory is further consolidated by animal studies that deliver cell-permeable PxIxIT and LxVP peptides to achieve immunosuppression with fewer side effects (*Elloumi et al., 2012*; *Escolano et al., 2014*; *Noguchi et al., 2004*).

In conclusion, the structural insights in this study offer fresh mechanistic perspective on how AKAP79 assemblies influence calcium signaling. Live-cell imaging experiments in *Figure 7* show that engagement of the LKIP sequence attenuates anchored phosphatase activity at low levels of calcium. Importantly, these effects are relieved upon increased calcium influx. Thus, proximity or direct association with L-type calcium channels, as reported by others, may profoundly influence the temporal action of the anchored phosphatase. This is particularly relevant in the cardiovascular system, where transient calcium fluctuation is pronounced within the vicinity of L-type calcium channel-AKAP79/150 clusters (*Nieves-Cintrón et al., 2016*; *Nystoriak et al., 2014*; *Nystoriak et al., 2017*). A similar situation may arise in the brain where neuronal NMDA receptors, L-type calcium channels in the post-synaptic density of excitatory synapses and GABA receptors embedded within inhibitory synapses are reversibly controlled by AKAP79/150 associated pools of PKA, PKC and PP2B (*Brandon et al., 2003*; *Colledge et al., 2000*; *Jurado et al., 2010*; *Li et al., 2012*; *Oliveria et al., 2007*; *Sanderson et al., 2012*). With this in mind, we postulate LKIP motifs on AKAP79/150 differentially regulate PP2B responses to calcium stimulation and hence balance the role of CaMKII phosphorylation at synapses (*Derkach et al., 1999*; *Mulkey et al., 1994*). Our discovery that flexibility is a defining feature of AKAP79 complexes points toward a mechanism whereby combinatorial recruitment of binding partners tailors the overall conformation of the macromolecular assembly. Such structural versatility represents a new means to shape AKAP79/150 kinase-phosphatase assemblies for their customized physiological roles.

## Materials and methods

### Bioinformatic predictions

For all predictions, the amino acid sequence of human AKAP79 was used. Disorder predictions were obtained using default parameters on IUPred (iupred.enzim.hu) and PONDR (pondr.com) (*Dosztányi et al., 2005*; *Li et al., 1999*). Predictions of short linear interaction motifs were obtained

using ANCHOR (anchor.enzim.hu) and SLiMPred (bioware.ucd.ie) (*Dosztányi et al., 2009*; *Mooney et al., 2012*). Graphs were prepared using Prism 6.0 (GraphPad Software).

## Protein expression and purification

All proteins were transformed and expressed in BL21 (DE3) pLysS cells (Life Technologies). AKAP79 was expressed as an MBP fusion in a modified pMAL c5x backbone (NEB). In addition, a 10x His-tag was placed at the C-terminus of the AKAP79 sequence. AKAP79-transformed BL21 cultures were grown in Terrific Broth (Sigma) until $OD_{600}$ = ~0.5, and expression was then induced with 0.4 mM IPTG at 37°C for 4 hr. Cells were then pelleted, frozen at −20°C and thawed and resuspended in 30 mL/liter of culture Buffer A (200 mM NaCl, 20 mM HEPES, pH 7.5) 4 mg/mL lysozyme, 1 mM AEBSF, 2 μg/mL leupeptin, 16 μg/mL benzamidine. After resuspension, Triton X-100 was added to 0.5%, and benzonase (Sigma-Aldrich) was added at a 1:20,000 dilution. After incubation for an additional 30 min, the lysate was cleared by spinning at 20,000 rpm in an SA-600 rotor (Sorvall) for 30 min. Clarified lysate was incubated with nickel affinity resin (Roche) for 1 hr, and then allowed to flow through by gravity. 2 mL wash/elution fractions were collected and analyzed by Coomassie staining containing the following concentrations of imidazole: 10, 20, 30, 50, 75, 100, 250, 500 mM (x5). Fractions containing MBP-AKAP79 were then combined and concentrated to <5 mL and applied to a HiLoad 16/600 Superdex 200 gel filtration column for separation at 0.5 mL/min. The peak corresponding to soluble MBP-AKAP79 was pooled and concentrated and flash-frozen, then stored at −80°C.

GST-PP2B was cloned into the pGEX-6P1 backbone as a bicistronic expression vector containing a Shine-Dalgarno sequence between the A subunit and the B subunit. Calmodulin was also expressed as a GST fusion in the pGEX-6P1 vector. PP2B and calmodulin were both purified by affinity chromatography, the GST tag was cleaved and gel filtration chromatography was performed using a HiLoad 16/600 Superdex 200 gel filtration column for PP2B and a HiLoad 16/600 Superdex 75 gel filtration column for calmodulin. PP2B purified for use in subunit labeling or GST pulldown assays included a Flag tag located on the C-terminus of the catalytic subunit.

Complexes were formed by incubating 1 mg of MBP-AKAP79 with PP2B and CaM in a 1:2.5:3 molar ratio overnight in buffer A supplemented with either 2 mM $CaCl_2$ or 2 mM EDTA. These samples were then injected onto to a Superdex Increase 200 10/300 column (GE) or a HiLoad Superdex 200 preparative column using an AKTApurifier FPLC in buffer A with $CaCl_2$ or EDTA. Peaks were analyzed by SDS-PAGE and Coomassie staining. The first peak, which elutes after the void volume, was pooled for further experimental analysis.

## Light scattering

The SEC-MALS system used was an AKTApure FPLC (GE), with an in-line Optilab T-Rex refractometer (Wyatt), and a Dawn Heleos II light scattering instrument (Wyatt). 500 μL of purified complex (~2 mg/mL) was injected onto a WTC-050S5 SEC column (Wyatt) and eluted at 0.5 mL/min directly into the on-line MALS instruments. Data was collected and processed to determine molecular mass using Astra 6 (Wyatt).

## Crosslinking and native PAGE

Samples were stabilized for native PAGE analysis by crosslinking with 250 μM BS3 (Thermo Fisher) for 30 min at room temperature. The crosslinking reaction was quenched by addition of Tris, pH 8.0 to 0.5 mM. The sample was then applied to NativePAGE 4–16% Bis-Tris gels (Life Tech.) according to the manufacturers' instructions.

## Western blotting

All western blots were performed by transferring samples from SDS-PAGE gels to nitrocellulose membranes at 1.00A for 36 min. Membranes were blocked in 5% milk in TBS/T plus 0.02% sodium azide. The following primary antibodies were used – mouse monoclonal anti-MBP-HRP 1:1000 (NEB), mouse monoclonal anti-FLAG-HRP 1:4000 (Sigma), rabbit polyclonal anti-AKAP79 C-terminus (Millipore), rabbit polyclonal anti-PP2B A subunit (Millipore), mouse monoclonal anti-PP2B B subunit (Abcam), and rabbit polyclonal anti-CaM (Santa Cruz). Blots were incubated in primary antibody dilutions overnight at 4°C, and then washed 3 times for 5 min in TBS/T. Blots were then incubated with

the appropriate secondary antibody conjugated to HRP at 1:10,000 dilution for 1 hr at room temperature. After three more TBS/T washes, blots were developed and imaged.

## GraFix preparation

In order to prepare AKAP79/PP2B/CaM complexes for EM analysis,~2 mg purified complex was concentrated to ~100 µL and applied to a continuous density/glutaraldehyde gradient. This gradient was prepared in 13.8 mL tubes, with 5–30% (w/v) glycerol and 0–0.15% glutaraldehyde (*Kastner et al., 2008*). Samples were spun in an SW41Ti rotor (Beckmann-Coulter) at 35,000 rpm for 18 hr at 4˚C. The gradients were fractionated in ~200 uL aliquots and analyzed by SDS-PAGE. Fractions that contained a single stabilized band were selected for further analysis.

## Negative stain grid preparation

For random conical tilt experiments, we used C-flat holey carbon support grids (Protochips, prod #CF-2/.5–4C) that were coated with a thin layer of carbon evaporated onto mica and then floated on ultrapure water. For other negative stain experiments, we used standard carbon support grids (Ted Pella, G-400) coated with carbon by evaporation. All grids were glow discharged and then ~5 µL of sample was allowed to adsorb to the grid for approximately 20 s. Grids were then blotted dry and 2% uranyl formate was added for 2 min. After blotting excess uranyl formate and allowing to dry, grids were ready to image.

## Random conical tilt data acquisition

Micrographs of untilted and −55˚ tilted views were acquired on a FEI T12 Spirit operated at 120 kV, spot size 5, 52000x nominal magnification, pixel size 2.07 Å, defocus values between −0.7 and −1.5 µm, and a dose of 30 e-/Å2. Data collection was automated using the MSI-RCT application within the Leginon software package (*Suloway et al., 2005*). For 2-D analysis of labeled complexes, data was collected using the MSI-Raster application in Leginon.

## Generation of Fab fragments and complex labeling

Fab fragments were generated from M2-anti-Flag mouse monoclonal (Sigma) and anti-MBP mouse monoclonal (NEB) antibodies by digestion with ficin or papain, respectively. After 6 hr digestion at 37˚C, digested material was incubated with protein-A agarose to capture Fc fragments and undigested whole IgG. Fab fragments were further purified using size-exclusion chromatography on a Superdex 200 gel filtration column. These fragments were then incubated with AKAP79/PP2B/CaM complexes overnight at 4˚C and subjected to an additional round of size exclusion chromatography. The elution fractions corresponding to labeled complexes were collected, pooled and subjected to GraFix preparation for negative stain EM.

## Labeling with gold particles

1.4 nm Nanogold particles (Nanoprobes) were incubated in molar excess with AKAP79/PP2B/CaM complexes overnight at 4˚C in buffer A supplemented with calcium. Subsequently, the labeled complexes were separated using a Superdex 200 column, and diluted to prepare for negative stain EM.

## EM data processing

Data was processed in the Appion pipeline with the following programs (*Lander et al., 2009*). Particles were picked using DoGPicker, and tilt-pairs were determined using AutoTiltPicker (*Voss et al., 2009*). The contrast transfer function was determined using CTFFind (*Mindell and Grigorieff, 2003*), and corrected using the EMAN 1.9 phase flip method. Individual particles were clustered using Xmipp 3 cl2d reference-free alignment to yield initial sets (*Sorzano et al., 2010*). After discarding classes with junk particles for several iterations, references were selected and used for reference-based alignment with the SPIDER AP MQ command (*Frank et al., 1996*). These references were then used to create RCT volumes using SPIDER within the Appion interface.

## Protein crosslinking, sample preparation, and mass spectrometry

Eluted proteins in 1 mL of 20 mM HEPES pH 8.5, 200 mM NaCl, and either 2 mM CaCl2 or 2 mM EDTA were crosslinked with 10 mM Biotin-Aspartate Proline-PIR n-hydroxyphthalimide (BDP-NHP)

(*Weisbrod et al., 2013*). As necessary, the pH was adjusted to ~8.0 with 100 μl of 200 mM HEPES pH 8.5. The reaction was allowed to continue for 1 hr at room temperature. Crosslinked proteins were denatured by the addition of urea buffer (8 M urea, 100 mM Tris-Cl pH 8.0), reduced (5 mM dithiothreitol, 30 min, 55°C), and alkylated (15 mM iodoacetamide, 1 hr, dark, room temperature). Crosslinked proteins were then digested with sequencing grade trypsin (Promega) overnight at 37°C. Resulting peptides were desalted with C18-SepPaks (Waters) and dried by vacuum centrifugation. Crosslinked peptides were injected onto an in-house pulled C8 column (3 μm, 200 Å, Magic) and analyzed by Real-time analysis of crosslinked peptide technology (ReACT) (*Weisbrod et al., 2013*). Spectra generated from ReACT were searched against a target-decoy database using SEQUEST (*Eng et al., 1994*). The complete set of observed peptides is presented in *Supplementary file 1* along with their Expect scores and PPM error for the best-scoring relationships observed for each peptide pair. Crosslinked sites were mapped to proteins using xiNet (*Combe et al., 2015*).

## Structural modeling

Crosslinked peptides on PP2B, the PP2B/cyclosporin complex, and the PP2B/LxVP complex were modeled using Pymol (Schrödinger). RCT volume data was modeled using Chimera (*Pettersen et al., 2004*), and crystal structures were fit in the maps using the Fit in Map command.

## Computational peptide/protein docking

The putative LxVP peptide in AKAP79 (KSRLKIPCIKFPRG) was computationally docked onto a crystallographic model of PP2B using the CABS-Dock server (*Blaszczyk et al., 2016*). The peptide was assumed to form a random loop (as suggested by previous predictions), and the known binding site for the PxIxIT motif was excluded from being a possible binding site. The lowest energy conformation was assumed to be the most accurate binding prediction. This was then used as the input for the Rosetta FlexPepDock server (*London et al., 2011*), which refined the results to produce the final displayed model.

## Solid phase peptide synthesis and overlay

Peptides were synthesized onto a cellulose membrane using the Intavis MultiPep solid-phase peptide synthesizer. After resolubilizing in ethanol, the peptides were overlaid with Flag-PP2B at ~1 mg/mL in 1% BSA/TBS-T in the presence of either 100 μg/mL CaM and 5 mM CaCl2 or 5 mM EDTA. After washes in TBS/T supplemented with calcium or EDTA, the dot blots were developed.

## GST protein-protein interaction assays

GST protein-protein interaction assays were performed using glutathione beads saturated with purified fragments of AKAP79. 2 μg Flag-PP2B was added to each sample, incubated for 2 hr, and then washed five times with RIPA buffer, followed by SDS-PAGE and Western blotting analysis. An LxVP peptide derived from NFATc1 or a scrambled control peptide was added to a final concentration of 200 μM during the incubation step. In some cases, a mutant form of Flag-PP2B was used, while in others, a pre-formed cyclosporin A/cyclophilin A complex was used as a competitor (2 μM). After the final wash, PP2B activity buffer (Promega) was added and a phosphopeptide substrate (Promega) was included to measure phosphatase activity in these samples.

## Alphascreen competition assays

Alphascreen competition assays were performed in 25 mM HEPES, pH 7.5, 100 mM NaCl, 0.1% BSA, 3 mM CaCl2, 2 μg/mL CaM. 10 μL of biotinylated PP2B and 10 μL of a His-tagged AKAP79 fragment (final concentration of each - 100 nM) were mixed and incubated for 15 min. 10 μL serial dilutions of untagged PP2B were added to the wells for 15 min and then Alphascreen beads (streptavidin donor, and nickel acceptor; Perkin-Elmer) were added and incubated for 60 min. Following this, the AlphaScreen signal was detected using a BMG PolarStar Omega plate reader. Data was analyzed using Prism 6.0 (GraphPad), and fit using a one-site IC50 model. Because of the concentrations used, the IC50 is able to approximate the Kd value of the interaction.

## Design of AKAP79-CaNAR reporters

AKAP79 was fused to the N-terminal of the CaNAR2 sequence in the pcDNA 3.1 backbone. Mutation of the PxIxIT motif in NFAT (PRIEIT) to PRAEAT was done to abolish direct PP2B binding. Fusing a short peptide from AKAP79 to the CaNAR2 sequence and testing whether this was capable of producing a FRET response to ionomycin confirmed the sufficiency of the AKAP79 PxIxIT motif. After fusing the full-length sequence of AKAP79 to CaNAR, mutants were made which were lacking residues 122–136, 337–343, or both.

## Confocal imaging of AKAP79-CaNAR mutants

HEK293 cells were seeded on 12 mm poly-D-lysine and laminin coated coverslips (Fisher) and transfected with 0.3 µg of the wild type and mutant CaNAR reporter constructs. After 48 hr, cells were fixed in 4% paraformaldehyde at room temperature for 10 min and permeabilized for 1 hr in PBS with 0.1% Triton X-100. Coverslips were blocked in PBS with 10% donkey serum for 2 hr at room temperature before primary antibody staining overnight with mouse anti-PP2B antibody (BD Biosciences). Samples were washed 3x and incubated with goat anti-mouse Alexafluor-555 secondary antibody. Nuclei were stained with DRAQ5 (Cell Signaling Technology) for 15 min at room temperature and then washed 3x before mounting in ProLong Gold anti-fade reagent (Invitrogen) onto glass microscope slides. Maximum projection images were acquired with a Zeiss scanning laser confocal microscope using a 63X oil immersion objective. HEK293 cells were obtained from GE Life Sciences (Cat. # HCL4517), maintained separately from other cells and were screened weekly to confirm the absence of mycoplasma contamination. As the origin of the cells was not central to the nature of these experiments, we did not further validate the identity of the HEK293 cells.

## Immunoprecipitation of AKAP79-CaNAR

2 µg of each AKAP79-CaNAR variant were transfected into HEK293 cells for 48 hr. Cells were lysed in IP buffer (0.5% NP-40, 100 mM NaCl, 50 mM Tris-HCl, pH 7.4) supplemented with protease inhibitors. After lysates were cleared, they were incubated with 2 µg of mouse anti-GFP (Life Technologies), and 25 µL of protein A/G agarose for 2 hr. Following this, the beads were washed in IP buffer 3x and then SDS sample buffer was added. The samples were run on SDS-PAGE and transferred to nitrocellulose for western blotting as described above. 15 µg of lysate was used as an input.

## FRET measurements in response to ionomycin, high and low levels

CaNAR2pax was generated by substituting the isoleucine residues at positions 115 and 117 within the NFAT domain of CaNAR2 (*Mehta et al., 2014*) with alanines (113-PRIEIT-118 to 113-PRAEAT-118) via site-directed mutagenesis, thereby eliminating the endogenous calcineurin-docking PxIxIT motif. AKAP-tethered CaNAR2pax constructs were subsequently generated by PCR-amplifying full-length wild-type AKAP79, AKAP79ΔLKIP, AKAP79ΔPIX, and AKAP79ΔLKIPΔPIX using HindIII/BamHI-linker primers and ligating the resulting PCR fragments into HindIII/BamHI-digested CaNAR2pax in pcDNA3, yielding AKAP79WT-CaNAR2pax, AKAP79ΔLKIP-CaNAR2pax, AKAP79ΔPIX-CaNAR2pax, and AKAP79ΔLKIPΔPIX-CaNAR2pax, respectively. All constructs were verified by sequencing.

HeLa cells were cultured in Dulbecco minimal Eagle Medium (Gibco) containing 1 g/L D-glucose and supplemented with 10% fetal bovine serum (Sigma) and 1% penicillin/streptomycin (Sigma-Aldrich). Cells were maintained at 37°C in a humidified incubator with 5% $CO_2$. Prior to imaging experiments, cells were plated onto sterile 35 mm glass-bottom dishes, transfected with the indicated biosensor constructs at 70–80% confluency using Lipofectamine 2000 (Invitrogen), and then grown for an additional 48 hr. HeLa cells were obtained from ATCC (Cat. #CCL-2), maintained separately from other cells and were screened weekly to confirm the absence of mycoplasma contamination. As the origin of the cells was not central to the nature of these experiments, we did not further validate the identity of the HeLa cells.

Cells were washed twice with Hank's Balanced Salt Solution (Gibco) supplemented with 20 mM HEPES, pH 7.4, and 2.0 g/L D-glucose, then imaged in the dark at 37°C. Ionomycin (Calbiochem) was prepared at a stock concentration of 1 mM in DMSO and directly added to imaging dishes at the indicated concentrations. Images were acquired on an Zeiss Axio Observer.Z1 microscope (Zeiss) equipped with a 40x/1.3 NA oil-immersion objective lens, a Definite Focus system (Zeiss), and an electron-multiplying cooled charge-coupled device camera (Roper Scientific) controlled by Metafluor

7.7 software (Molecular Devices). Dual emission ratio imaging was performed using a 420DF20 excitation filter, a 450DRLP dichroic mirror, and two emission filters (475DF40 for CFP and 535DF25 for YFP). Filter sets were alternated using a Lambda 10–2 filter changer (Sutter Instruments). Exposure times were 50–500 ms, and images were acquired every 30 s.

## Quantification and statistical analysis
### Class average dimensional analysis
Individual classes were measured in ImageJ and binned into 1 nm groups for histogram analysis. Histograms were fit with Gaussian curves or the sum of two Gaussian curves in Prism (GraphPad). Isosurface thresholds of selected 3-D models were chosen to match dimensions of 2-D classes, and measurements of these model dimensions was carried out with Chimera (UCSF).

### AlphaScreen competition assays
AlphaScreen intensity values were normalized and then fit using the one site-fit log IC50 model in Prism (GraphPad). Experiments were carried out with four technical replicates.

### CaNAR response and kinetic analysis
Fluorescence was quantified in each channel by calculating the average fluorescence intensity in a manually defined region of interest (ROI). ROIs were drawn around individual cells displaying clear, plasma membrane-localized fluorescence, defined by uniform fluorescence across the cell surface with no nuclear shadow or fluorescence from intracellular membranes, along with a highlighted cell border or the appearance of membrane ruffles/protrusions. Background correction of the fluorescence images was performed by subtracting the intensities of un-transfected cells or regions of the imaging dish with no cells. Time-courses were normalized by setting the pre-treatment emission ratio as equal to one. Graphs were plotted using Prism 6.0 (GraphPad), and statistical analyses were performed using the same software. Statistical significance was set at $p < 0.05$.

### Phosphatase activity assays
Free phosphate was measured using a colorimetric assay (Promega), and the average of four experiments ± SEM was determined using Prism (GraphPad) by normalizing to the highest response in each experiment.

## Acknowledgements
We thank members of the Scott Lab for critical feedback, and M Milnes for administrative support.

## Additional information

### Funding

| Funder | Grant reference number | Author |
| --- | --- | --- |
| Howard Hughes Medical Institute | | John D Scott |
| National Institutes of Health | 1R01GM120553 | David Veesler |
| National Institutes of Health | 5R01DK105542 | John D Scott |
| National Institutes of Health | 4P01DK054441 | John D Scott |
| National Institutes of Health | R01DK073368 | Jin Zhang |

The funders had no role in study design, data collection and interpretation, or the decision to submit the work for publication.

### Author contributions
Patrick J Nygren, Conceptualization, Formal analysis, Investigation, Visualization, Writing—original draft, Writing—review and editing; Sohum Mehta, Devin K Schweppe, Jennifer L Whiting, Formal

analysis, Investigation, Writing—review and editing; Lorene K Langeberg, Investigation, Visualization, Writing—review and editing; Chad R Weisbrod, Investigation, Methodology, Writing—review and editing; James E Bruce, Jin Zhang, David Veesler, Conceptualization, Resources, Supervision, Funding acquisition, Writing—review and editing; John D Scott, Conceptualization, Resources, Supervision, Funding acquisition, Investigation, Visualization, Writing—original draft, Project administration, Writing—review and editing

## Author ORCIDs
Patrick J Nygren ⓘ http://orcid.org/0000-0001-8688-9389
Lorene K Langeberg ⓘ http://orcid.org/0000-0002-3760-7813
Jin Zhang ⓘ http://orcid.org/0000-0001-7145-7823
John D Scott ⓘ http://orcid.org/0000-0002-0367-8146

## Decision letter and Author response
Decision letter https://doi.org/10.7554/eLife.30872.021
Author response https://doi.org/10.7554/eLife.30872.022

## Additional files

### Supplementary files
• Supplementary file 1. (Related to *Figure 5*) Complete table of non-redundant observed crosslinked peptides in the presence of $Ca^{2+}$ or EDTA.
DOI: https://doi.org/10.7554/eLife.30872.019

• Transparent reporting form
DOI: https://doi.org/10.7554/eLife.30872.020

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
