## [Decision Letter]

Thank you for submitting your article "Intrinsic disorder within AKAP79 fine-tunes anchored phosphatase activity toward substrates and drug sensitivity" for consideration by *eLife*. Your article has been reviewed by three peer reviewers, and the evaluation has been overseen by a Reviewing Editor and Ivan Dikic as the Senior Editor. The reviewers have opted to remain anonymous.

The reviewers have discussed the reviews with one another and the Reviewing Editor has drafted this decision to help you prepare a revised submission.

Summary:

Nygren et al. describe a combination of negative stain EM studies, crosslinking studies, and in vitro plus cellular biochemistry studies to analyze Ca^2+^ regulation of AKAP79/PP2B complexes (when CaM binds). The EM data provide a strong case that active CaM/MBP-AKAP79/PP2B complexes are more compact/globular than inactive MBP-AKAP79/PP2B complexes – with max dimension reduced to 178Å from 184-228Å. The subunit mapping data are also interesting, and are interpreted in a reasonable way for the resolution of the study. The EM studies are nicely bolstered by the crosslinking studies shown in Figure 5, which make a compelling case (strengthened further by the data in Figure 6) that additional AKAP79/PP2B B and AKAP79/PP2B A (and PP2B B/A) interactions are acquired in the active complex that may influence the Ca++ sensitivity of the complex (Figure 7).

Overall the work is generally of high quality and provides new insight into the structural features of AKAP79, revealing a new short linear motif in AKAP79 that appears to functionally resemble the LxVP motif already known to provide a targeting function for PP2B substrates.

Essential revisions:

1) There are additional (weaker) bands in the SDS PAGE profiles in Figure 1 and more notably Figure 1. In the latter, there are 3 bands, about 25% of the intensity of PP2B A, at 75 kDa, 45 kDa, and 30 kDa. What are these, and why do they migrate in SEC with the MBP-AKAP79/PP2B/CaM complex? The authors should mention/discuss these. Are they co-assembled proteolytic fragments, for example? They are not seen in Figure 1, so may form during manipulations? The extra bands are also evident in Figure 2. Can the authors be sure that these are not coming from contaminating particles that could cause affect EM particle picking? This is particularly important given that the length distribution is bimodal in Figure 3. Could there be more than one covalent species?

2) It is puzzling that the EM data suggest that the active CaM/MBP-AKAP79/PP2B complex is more compact than the inactive MBP-AKAP79/PP2B complex, yet the former elutes slightly earlier from the SEC column in Figure 2/C (which would suggest the opposite). The authors should discuss this discrepancy. If the AKAP79-PP2B holoenzyme complexes exist in a variety of flexible and extended topologies, should this not be reflected in an earlier elution in SEC studies than the active complex?

3) A very basic summary of crosslinking/mass spectrometry data is presented in Supplementary file 1, but it is difficult to evaluate the information in any quantitative fashion. From a list of ~100 crosslinked peptides in either of the inactive and active complexes, 5 paired peptides are selected for discussion, with presumably several intramolecular interactions being summarized by the light grey semi-circles in Figure 5. What were the criteria used to filter the data. Is there some way to assess the relative abundance of any paired interaction versus the others. In Supplementary file 1, there are repeated entries for some pairs of peptides. What does this mean? There are apparent interactions of the LKIPCIKFPR peptide with a GAWASLKR peptide in the dormant complex, but there seem to be qualitative changes in the interactions of the LKIPCIKFPR peptide with the GAWASLKR peptide and others in the active complex.

4) The authors include images/FRET movies for AKAP79-CaNAR and AKAP79delL-CaNAR in Figure 7, but not for the δ P version or lower ionomycin concentrations? Similarly, time course data (Figure 7) are provided only for the high ionomycin concentration, not for the 100nM ionomycin studies. Inclusion of such additional figures would make the data more convincing. It is also not clear from the methods how the cells used for the analysis in Figure 7 were selected. It is stated that only cells with clear PM-localized fluorescence were used. How was this discerned? This issue should be explained more clearly in the methods. Also, how do the kinetics vary with expression level (if they do at all)?

5) The in vivo FRET analysis is interpreted to show that the LKIP motif decreases Ca++ sensitivity (Figure 7). The mechanistic basis for the decrease in Ca++ sensitivity is unclear. Is this because the LKIP motif competes with LxVP interaction motifs on substrates? If this is the mechanism, then this would not be a change in Ca++ sensitivity, as stated (?). Does the presence of a LxVP motif in the substrate affect activity of PP2B bound to AKAP or AKAP∆L complexes?

---

## [Author Response]

Essential revisions:1) There are additional (weaker) bands in the SDS PAGE profiles in Figure 1 and more notably Figure 1. In the latter, there are 3 bands, about 25% of the intensity of PP2B A, at 75 kDa, 45 kDa, and 30 kDa. What are these, and why do they migrate in SEC with the MBP-AKAP79/PP2B/CaM complex? The authors should mention/discuss these. Are they co-assembled proteolytic fragments, for example? They are not seen in Figure 1, so may form during manipulations? The extra bands are also evident in Figure 2. Can the authors be sure that these are not coming from contaminating particles that could cause affect EM particle picking? This is particularly important given that the length distribution is bimodal in Figure 3. Could there be more than one covalent species?

Previous work from our laboratory (Gold et al., 2011) analyzed the AKAP79-PP2B-CaM complex by mass spectroscopy (MS). Using this sensitive approach, the additional bands at 45 kDa and 30 kDa were assigned as PP2B degradation products. The band at 75 kDa is the result of the loss of the MBP tag on AKAP79. We now cite the Gold et al. reference in the Results section of the revised manuscript.

The GraFix treatment results in stabilization of a single species at a consistent molecular weight (Figure 2), indicating that this preparatory step results in further purification, excluding these minor contaminants. We include clarification of this point in subsection “Conformational variability in AKAP79/PP2B assemblies “in the revised manuscript.

2) It is puzzling that the EM data suggest that the active CaM/MBP-AKAP79/PP2B complex is more compact than the inactive MBP-AKAP79/PP2B complex, yet the former elutes slightly earlier from the SEC column in Figure 2/C (which would suggest the opposite). The authors should discuss this discrepancy. If the AKAP79-PP2B holoenzyme complexes exist in a variety of flexible and extended topologies, should this not be reflected in an earlier elution in SEC studies than the active complex?

This is a valid point. As the reviewer noted, we observe slightly different retention times of the complexes when purified using a HiLoad Superdex 200 SEC column. This may be explained by the presence of an additional component (CaM) that is incorporated into the active complex. However, these experiments were conducted on preparatory columns rather than analytical columns. Consequently, retention times varied from experiment to experiment. For this reason, we were uncomfortable drawing any firm conclusions regarding the slightly different SEC profiles of the active and dormant complexes. This information has been added in subsection “Conformational variability in AKAP79/PP2B assemblies” of the revised manuscript.

Nonetheless, and in keeping with the reviewer’s observation, we noted that the peak width (measured at half of the peak maximum) of the dormant complex was broader than that of the active complex. This may indicate structural heterogeneity within the dormant complex.

3) A very basic summary of crosslinking/mass spectrometry data is presented in Supplementary file 1, but it is difficult to evaluate the information in any quantitative fashion. From a list of ~100 crosslinked peptides in either of the inactive and active complexes, 5 paired peptides are selected for discussion, with presumably several intramolecular interactions being summarized by the light grey semi-circles in Figure 5. What were the criteria used to filter the data. Is there some way to assess the relative abundance of any paired interaction versus the others. In Supplementary file 1, there are repeated entries for some pairs of peptides. What does this mean? There are apparent interactions of the LKIPCIKFPR peptide with a GAWASLKR peptide in the dormant complex, but there seem to be qualitative changes in the interactions of the LKIPCIKFPR peptide with the GAWASLKR peptide and others in the active complex.

As the reviewer notes, we discuss 5 paired peptides. These peptides represent the entirety of assigned intermolecular interactions between AKAP79 and PP2B subunits. We have clarified the basis for this selection in subsection “New intermolecular contacts between AKAP79 and active PP2B” of the revised manuscript.

The reviewer points out that many intramolecular interactions were observed, and in the scope of Figure 5, these interactions are represented by grey semi-circles. Each bar represents an individual protein sequence, and lines between different bars represent identified intermolecular interactions. The cross-linked mass spectrometry was carried out as a qualitative experiment (attempting to identify as many links as possible in both the active and inactive states), therefore we do not believe we can accurately make relative quantitative assessments of specific cross-links or compare the abundance of certain links in relation to other links.

Within the scope of the current experiments it is not possible to calculate the relative abundances of different peptide-pairs due to differential “flyability” of peptides. Individual peptide-pairs exhibit different chemical and physical properties (notably ionization efficiency and fragment efficiency), therefore we can observe varied spectral intensities or identification rates even when peptides are present in equimolar amounts. Repeated entries in the EDTA data were due to redundant cross-linked relationships. This redundancy explains the unclear relationships between the LKIPCKIFPR and GAWASLKR peptides in the dormant and active complexes. We have removed redundant observations in Supplementary file 1, and have included expect scores and PPM errors for the best scoring relationship observed in both the active and dormant states to address this confusion. A description of this has been added to the Materials and methods section.

Although spectral counting has been used to quantitatively assess cross-linking/MS experiments, the current experiment was designed as a qualitative comparison of two states for these proteins in complex and we believe that retrospective quantitative analysis would not be accurate.

4) The authors include images/FRET movies for AKAP79-CaNAR and AKAP79delL-CaNAR in Figure 7, but not for the δ P version or lower ionomycin concentrations? Similarly, time course data (Figure 7) are provided only for the high ionomycin concentration, not for the 100nM ionomycin studies. Inclusion of such additional figures would make the data more convincing. It is also not clear from the methods how the cells used for the analysis in Figure 7 were selected. It is stated that only cells with clear PM-localized fluorescence were used. How was this discerned? This issue should be explained more clearly in the methods. Also, how do the kinetics vary with expression level (if they do at all)?

We have included additional data in Figure 7 as requested. In addition, we have included supplemental movie files for these images.

For selection of cells included in the analysis, regions of interest (ROI) were drawn around individual cells that displayed clear, plasma membrane-localized fluorescence. This was defined by uniform fluorescence across the cell surface with no nuclear shadow or fluorescence from intracellular membranes, along with a highlighted cell border or the appearance of membrane ruffles/protrusions. We have included details regarding the methodology in subsection “CaNAR response and kinetic analysis”.

The kinetics do not vary with expression level (R-squared values of less than 0.1). We have included a discussion of this in subsection “The LKIP motif controls PP2B sensitivity to calcium/calmodulin”, and a supplemental figure demonstrating that these effects are not due to changes in expression (Figure S7).

5) The in vivo FRET analysis is interpreted to show that the LKIP motif decreases Ca++ sensitivity (Figure 7). The mechanistic basis for the decrease in Ca++ sensitivity is unclear. Is this because the LKIP motif competes with LxVP interaction motifs on substrates? If this is the mechanism, then this would not be a change in Ca++ sensitivity, as stated (?). Does the presence of a LxVP motif in the substrate affect activity of PP2B bound to AKAP or AKAP∆L complexes?

It is important to note that we used a modified version of the CaNAR reporter that is fused to AKAP79. Modifications included removal of PP2B anchoring motifs (LxVP or PxIxIT) from the CaNAR portion of the original reporter. The reason for such modifications was to ensure that recruitment of PP2B has to proceed through the AKAP79 portion of the modified phosphatase activity biosensor, and thereby eliminates the possibility of LxVP competition. Thus, we can conclude that upon Ca^2+^/CaM influx, engagement of the AKAP79 LKIP motif stabilizes a conformation of PP2B that does not relieve auto-inhibition as readily as non-anchored PP2B. We have more clearly emphasized this point in subsection “CaNAR response and kinetic analysis”.